# The structural basis of odorant recognition in insect olfactory receptors

Josefina del Mármol[1], Mackenzie A. Yedlin[1] & Vanessa Ruta[1✉]

Olfactory systems must detect and discriminate amongst an enormous variety of odorants[1]. To contend with this challenge, diverse species have converged on a common strategy in which odorant identity is encoded through the combinatorial activation of large families of olfactory receptors[1–3], thus allowing a finite number of receptors to detect a vast chemical world. Here we offer structural and mechanistic insight into how an individual olfactory receptor can flexibly recognize diverse odorants. We show that the olfactory receptor *Mh*OR5 from the jumping bristletail[4] *Machilis hrabei* assembles as a homotetrameric odorant-gated ion channel with broad chemical tuning. Using cryo-electron microscopy, we elucidated the structure of *Mh*OR5 in multiple gating states, alone and in complex with two of its agonists—the odorant eugenol and the insect repellent DEET. Both ligands are recognized through distributed hydrophobic interactions within the same geometrically simple binding pocket located in the transmembrane region of each subunit, suggesting a structural logic for the promiscuous chemical sensitivity of this receptor. Mutation of individual residues lining the binding pocket predictably altered the sensitivity of *Mh*OR5 to eugenol and DEET and broadly reconfigured the receptor's tuning. Together, our data support a model in which diverse odorants share the same structural determinants for binding, shedding light on the molecular recognition mechanisms that ultimately endow the olfactory system with its immense discriminatory capacity.

The olfactory system faces a unique challenge amongst sensory modalities owing to the inordinate complexity of the chemical world. Whereas light waves vary continuously in amplitude and frequency, odorants differ discretely along an enormous number of dimensions in their molecular structure and physicochemical properties. Consequently, just three photoreceptors are sufficient to sense the entire spectrum of visible light, but large repertoires of olfactory receptors appear to be necessary to detect and discriminate amongst the diversity of chemicals in the environment[1–3]. In mammals, odour detection is mediated by G-protein-coupled receptors that signal through canonical second-messenger cascades[5,6]. By contrast, insects detect volatile chemicals using a unique class of odorant-gated ion channels[7,8] consisting of two subunits: a conserved co-receptor (Orco) subunit[9,10] and a highly divergent odorant receptor (OR) subunit that contains the odorant-binding site and confers chemical sensitivity to the heteromeric complex[11].

Although mammals and insects rely on distinct molecular mechanisms for odour detection, they share a common neural logic for olfactory perception based on the combinatorial activation of distinct ensembles of olfactory receptors and associated sensory neurons[1,3,12]. Central to this sensory coding strategy is that most individual ORs detect a variety of structurally and chemically diverse odorants[11,13,17,25]. However, in the absence of a structural model, how such flexible chemical recognition is achieved has remained unknown. Whether the broad chemical tuning of ORs reflects the presence of multiple odorant-binding sites that differ in their chemical specificity or a single promiscuous binding pocket is not known. Furthermore, which

structural or chemical features of odorants are recognized by a receptor remains unclear. In this study, we leveraged the evolutionary diversification of insect ORs to elucidate the structures of a homomeric receptor from a basal insect species bound to different ligands. We reveal how a single receptor can detect a wide array of odorants through a single promiscuous binding site that recognizes the overall physicochemical properties of each odorant rather than being tuned to any of their specific structural or molecular features, suggesting a structural basis for flexible chemical recognition.

## *Mh*OR5 is a broadly tuned receptor

Although neopteran insects each express a repertoire of highly divergent ORs along with a single, almost invariant Orco[2], recent genomic analyses have revealed that some basal insects, such as the jumping bristletail *M. hrabei*, possess only a small number of OR genes and lack an apparent Orco orthologue[4] (Fig. 1a). *Mh*ORs have been proposed to represent the most ancestral members of the insect olfactory receptor family, arising before the emergence of Orco[4,14]. Although little is known about chemosensory detection in the jumping bristletail, we reasoned that *Mh*ORs might function as homomeric olfactory receptors. We heterologously expressed each *Mh*OR in HEK293 cells and found that, indeed, *Mh*OR1 and *Mh*OR5 migrated as tetramers on non-denaturing native gels (Extended Data Fig. 1a, b). To assess whether these homomeric complexes function as chemoreceptors, we used a high-throughput fluorescence assay[10] in which we co-expressed *Mh*OR1 or *Mh*OR5 with the indicator GCaMP6s

[1]Laboratory of Neurophysiology and Behavior, The Rockefeller University, New York, NY, USA. ✉e-mail: ruta@rockefeller.edu

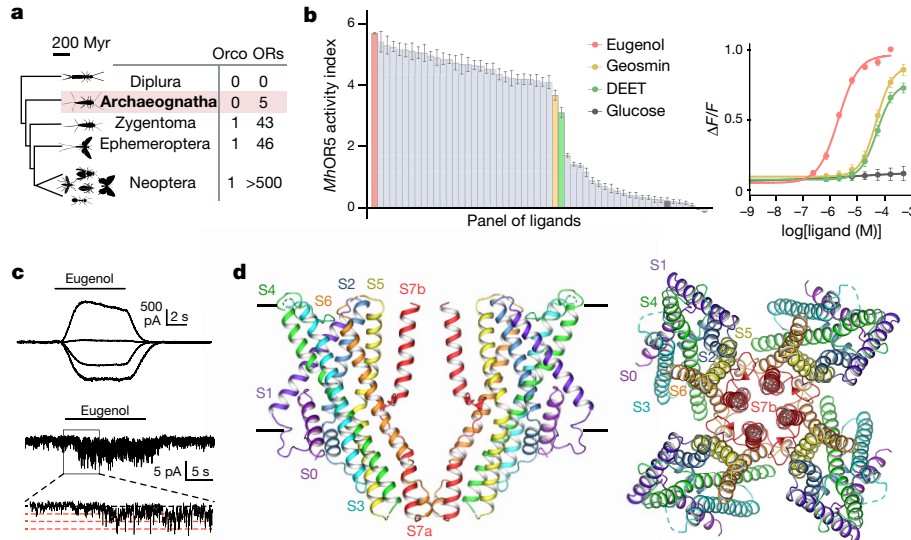

**Fig. 1 | The structure of _Mh_OR5, a broadly tuned odorant-gated ion channel.**
**a**, Phylogenetic tree of selected insect clades and the representative numbers of OR and Orco genes in their genomes. Myr, million years. **b**, Left, activity of _Mh_OR5 evoked by a panel of 54 small molecule ligands. Right, dose–response curves of _Mh_OR5 to selected ligands. For dose–response curves for all ligands, see Extended Data Fig. 2; _n_ values for biological replicates are in Supplementary Table 1. See Methods for a detailed description of the activity index. **c**, Representative traces of eugenol-evoked currents in HEK cells expressing _Mh_OR5. Top, whole-cell currents at voltages from −80 mV to 40 mV. Bottom, single-channel recordings in outside-out patches at −80 mV. Red dashed lines indicate current levels when different numbers of channels open. **d**, Cryo-EM structure of _Mh_OR5 shown from the side (left) and top (right). Subunits are coloured in rainbow palette from the N terminus (purple) to the C terminus (red). In the side view, front and back subunits were removed for clarity of visualization. Black lines, membrane boundaries.

and measured calcium influx in response to a panel of 54 small molecules over a range of concentrations. We found that _Mh_OR5 was activated by many volatile odorants but not tastants, consistent with a role for this receptor in olfactory detection (Fig. 1b, Extended Data Fig. 2a–d). _Mh_OR5 was also activated by the insect repellent DEET and inhibited by the synthetic Orco agonist VUAA1[15]. To quantitatively capture the complexity of odorant-evoked responses[16] (Extended Data Fig. 2a–d), we defined an activity index for each odorant ($-\log(EC_{50}) \times \max \Delta F/F$; in which $EC_{50}$ is the concentration of ligand at which the response reaches half its maximal value) that reflects both the apparent affinity and maximal efficacy of an odorant. _Mh_OR5 was activated by over 65% of odorants, resembling the broad molecular receptive fields of many insect and mammalian ORs[11,13,17] (Extended Data Fig. 1d). By contrast, _Mh_OR1 exhibited far more selective tuning, responding to only eight odorants from the same chemical panel (Extended Data Fig. 1e). Both _Mh_OR1 and _Mh_OR5 were activated by ligands that spanned multiple chemical classes and a range of physicochemical properties (Extended Data Fig. 1e, f), exemplifying the complex chemical logic of odorant detection.

Whole-cell recordings of HEK cells expressing _Mh_OR5 showed that the odorant eugenol elicited slowly activating inward currents that reversed at 0 mV, consistent with its function as a cation-selective ion channel (Fig. 1c). In outside-out patches, eugenol evoked small-conductance single-channel activity that rapidly flickered between the closed and open states, resembling canonical heteromeric insect olfactory receptors[7,10] (Fig. 1c, Extended Data Fig. 2e). _Mh_ORs thus autonomously assemble as homotetrameric odorant-gated ion channels and display the divergent chemical tuning profiles typical of this receptor family. Given its sensitivity to a broad array of structurally diverse odorants, we focused on _Mh_OR5 to investigate the molecular basis of promiscuous chemical recognition.

## Structure of the _Mh_OR5 homotetramer

We used single-particle cryo-electron microscopy (cryo-EM) to elucidate the structure of the _Mh_OR5 tetramer. We obtained a density map at 3.3 Å resolution (Fig. 1d, Extended Data Figs. 3, 4, Extended Data Table 1), which allowed us to unambiguously build a model for the majority of

the protein, with the exception of several extra-membranous loops and the short intracellular N terminus and extracellular C terminus (Extended Data Figs. 4c, 5c). A comparison of the structure of _Mh_OR5 with the previously elucidated structure of Orco from the wasp _Apocrypta bakeri_[10] showed that these two receptors, despite sharing only about 18% amino acid conservation, display notable similarity, both in the fold of each heptahelical subunit and in the tetrameric organization of the subunits within the membrane plane (Extended Data Fig. 5a, b). As in Orco, each _Mh_OR5 subunit contributes a single helix (S7b) to the central ion conduction pathway, and their S0–S6 helices form a loosely packed domain that projects radially away from the pore axis (Fig. 1d). Within the membrane, the contacts between _Mh_OR5 subunits are minimal and confined to the pore, whereas about 75% of the residues that form inter-subunit interactions reside within the intracellular 'anchor' domain, formed from the intertwined S4–S7 helices of all four subunits (Extended Data Fig. 5d). Analogous to the Orco structure, the tightly packed anchor domain of _Mh_OR5 exhibited the highest local resolution (Extended Data Fig. 4c), consistent with a structural role in stabilizing the loosely assembled S0–S6 transmembrane domains within the lipid bilayer. The limited sequence conservation across neopteran ORs and Orcos maps to residues predominantly within the pore and anchor domain[10], further underscoring how the architecture of this receptor family can accommodate a high degree of sequence diversification while maintaining the same overall fold, a feature that is likely to have facilitated the rapid evolution of ORs.

## Odorant binding leads to pore opening

To explore the structural determinants of odorant gating, we determined a 2.9 Å-resolution structure of _Mh_OR5 in complex with its highest activity ligand, eugenol. Three-dimensional reconstruction of the bound structure immediately yielded higher resolution, as was apparent from early stages of data processing (Extended Data Figs. 3, 4, Extended Data Table 1). The _Mh_OR5 pore displays the same distinct quadrivial architecture as the Orco homotetramer[10], in which a single extracellular pathway opens into a large aqueous vestibule near the intracellular surface of the

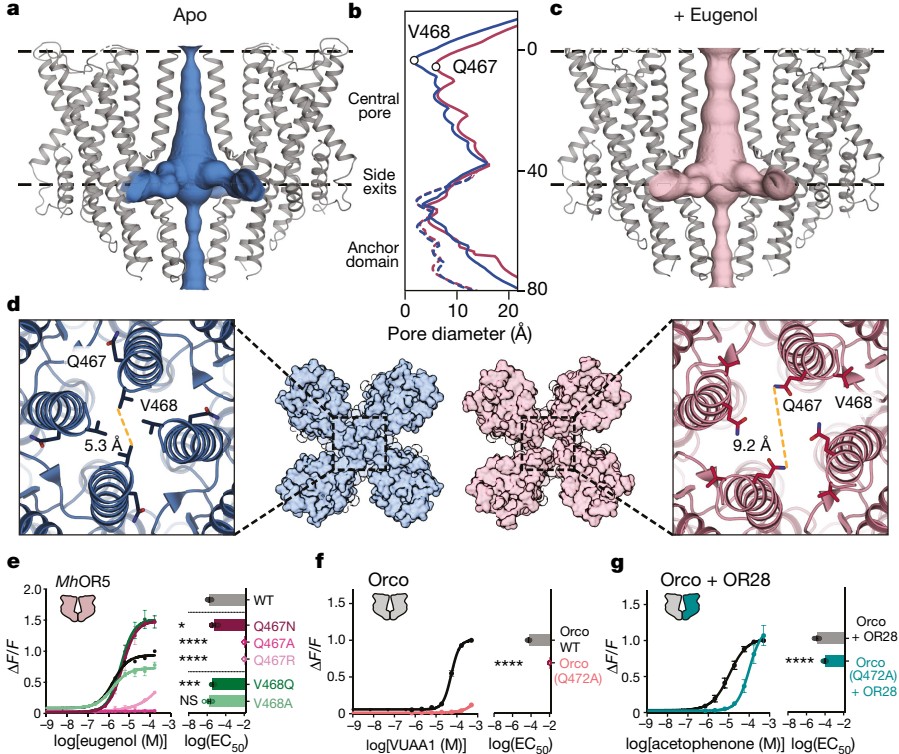

**Fig. 2 | Odorant-evoked opening of *Mh*OR5. a, c**, The channel pores of the apo (**a**, blue) and eugenol-bound *Mh*OR5 (**c**, pink). Black dashed lines, membrane boundaries. **b**, The diameter of the ion conduction pathway (solid lines) and along the anchor domain (dashed lines). *y*-axis shows distance from the outer membrane boundary towards the intracellular space in Å. **d**, Close-up view of the pore from the extracellular side in the apo (left, blue) and eugenol-bound (right, pink) structures. Distances between S7b residues measured from atom centres. **e–g**, Dose–response curves (left) and mean ± s.e.m. log(EC$_{50}$) (right) for wild-type (WT), Val468 mutants and Gln467 mutants of *Mh*OR5 (**e**),

wild-type and Q472A mutant (homologous to Gln467 in *Mh*OR5) of Orco as a homotetramer (**f**) and wild-type and Q472A mutant of Orco in heteromeric Orco–*Ag*OR28 complex (**g**). Statistical significance determined using one-way ANOVA followed by Dunnett's multiple comparison tests. For mutants for which the EC$_{50}$ was incalculably high and Bartlett's test showed non-homogenous variance, statistical significance determined with a Brown–Forsythe test. ****$P < 0.0001$; ***$P < 0.001$; *$P < 0.01$; NS, not significant. Supplementary Tables 2, 3 contain details including *n* values for all biological replicates.

membrane and then diverges into four lateral conduits formed at the interfaces between subunits (Fig. 2a, b). In the apo structure, the S7b helices coalesce to form the narrowest portion of the ion conduction pathway. In particular, Val468 protrudes into the channel lumen, generating a hydrophobic constriction measuring about 5.3 Å in diameter, and thus serves as a gate to impede the flow of hydrated ions through the quadrivial pore (Fig. 2a, b, d). In the presence of eugenol, the extracellular aperture of the pore is dilated as a result of movement of the S7b helices away from the central pore axis (Fig. 2b–d), which rotates Val468 out of the pore lumen to face the lipid bilayer, while Gln467 rotates in to face the ion pathway. As a consequence of this rearrangement, the chemical environment of the pore is transformed from a narrow hydrophobic constriction to a wide hydrophilic ring, 9.2 Å in diameter, that can readily accommodate the passage of hydrated cations. Notably, the remainder of the quadrivial pore remains essentially unaltered with the addition of eugenol (Fig. 2a–c), confirming that the tightly packed anchor domain forms a relatively stationary structural element[10]. The dilation of the S7b helices thus appears to be sufficient to gate the ion conduction pathway—this small conformational change would present a low energetic barrier to gating, consistent with the low affinity of most odorants[11,17] and with functional evidence that *Mh*OR5 channels, as with many insect olfactory receptors, open spontaneously even in the absence of ligand[7,11] (Extended Data Fig. 1c).

Gln467 is highly conserved across Orcos and ORs from *M. hrabei* and other basal insect species[14] and was previously identified as a component of the only signature sequence motif (TYhhhhhQF, in which h is any hydrophobic amino acid) that is diagnostic of the highly divergent

insect chemosensory receptor superfamily[18]. Mutation of Gln467 in *Mh*OR5 to either the smaller residue alanine or the positively charged residue arginine strongly impaired receptor function, whereas a more conservative mutation to asparagine had little effect on activity (Fig. 2e). Replacement of the neighbouring residue Val468 with either alanine or glutamine resulted in minimal changes to odorant activation (Fig. 2e), highlighting that movement of the pore helices, rather than simply the presence of a pore-lining glutamine, is necessary to gate the channel. In the closed structure of the Orco homotetramer[10], the homologous residue, Gln472, points into the lipid membrane, similar to its position in the closed conformation of *Mh*OR5. Mutation of Gln472 to alanine in Orco yielded non-functional homomeric channels (Fig. 2f). Gln472 is thus one of the few S7b residues in Orco that is intolerant to such a perturbation[10], consistent with a conserved and critical role for this residue in gating and/or ion permeation across this receptor family. Notably, the Q472A Orco mutant could be partially rescued by co-expression with an OR from *Anopheles gambiae* (Fig. 2g), indicating that this mutant can fold and function in the context of the heteromeric assembly and underscoring the intrinsic robustness of the Orco–OR complex, where both subunits contribute to a shared ion conduction pathway[10,19].

## Architecture of the odorant-binding site

In the transmembrane domain of each *Mh*OR5 subunit, the S2, S3, S4 and S6 helices splay apart to form a 15 Å-deep pocket within the extracellular leaflet of the bilayer (Fig. 3a, Extended Data Figs. 6, 7). Clearly defined density consistent with the size and shape of eugenol

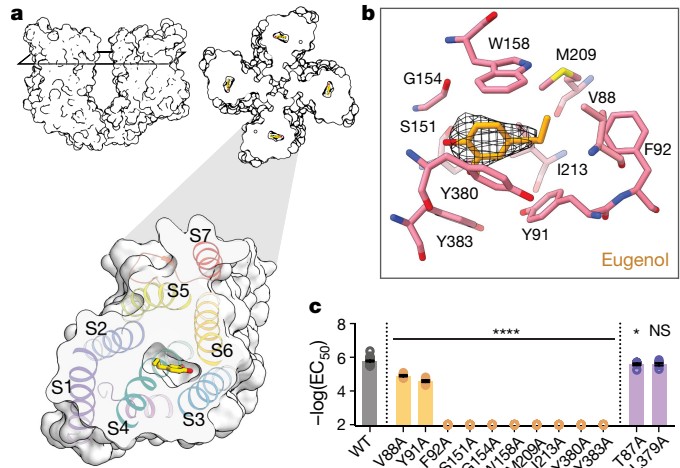

**Fig. 3 | Architecture of the odorant-binding site in *Mh*OR5. a**, Left, side view of two subunits of *Mh*OR5 shown as a surface representation with the cross-section through the binding pocket indicated. Right, top view of a cross-section of the *Mh*OR5 tetramer through the binding pocket. Bottom, expanded view of a single subunit shown as a surface representation with helices coloured and labelled. Eugenol shown in stick representation. **b**, View of the binding pocket. Pink, residues in contact with eugenol; yellow, eugenol; black mesh, cryo-EM density for eugenol. **c**, Mutagenesis of residues in contact with eugenol (gold) and two neighbouring residues (purple) that project away from the pocket. Mean ± s.e.m. log(EC$_{50}$) shown. Statistical significance determined using one-way ANOVA followed by Dunnett's multiple comparison tests. For mutants for which the EC$_{50}$ was incalculably high and Bartlett's test showed non-homogenous variance, statistical significance determined with a Brown–Forsythe test. ****$P < 0.0001$; *$P < 0.05$; NS, not significant. Dose–response curves shown in Extended Data Fig. 9, additional views of the pocket in Extended Data Figs. 6, 8, 9d, and *n* values for all biological replicates in Supplementary Table 2.

lies at the base of this pocket, enclosed within a hydrophobic box constructed from several large aromatic and hydrophobic residues, with Trp158 forming the lid, Tyr91 and Tyr383 forming its base, and flanked by Tyr380 on one side and by Met209, Ile213 and Phe92 on the other (Fig. 3b, Extended Data Figs. 6, 9d). In the apo structure, the density for some of these amino acids was diffuse (Extended Data Fig. 6b), which could be attributed to the overall lower resolution of this structure or to conformational flexibility when no odorant is bound. The lower resolution of the apo pocket precluded us from defining the path that eugenol takes to enter the pocket, as in the bound structure the pocket is not obviously accessible to solvent (Extended Data Figs. 6b, 7a), or from determining whether the cavity is partially occupied in the absence of an added ligand. Binding of eugenol, however, stabilized the residues that line the pocket, allowing unambiguous mapping of the side chains that form the binding site.

To explore the potential binding modes of eugenol, we used computational docking methods[20] and performed a broad grid search spanning the majority of the transmembrane domain. This analysis identified a series of closely related eugenol poses with uniformly favourable docking scores that fit into the experimental density well (Extended Data Fig. 8a). At this resolution, differentiating between these poses is challenging given that eugenol, as with most odorants, is a small molecule with few distinguishing structural features to orient it within the density. Nevertheless, eugenol was predicted to bind through comparable interactions across all the top poses, but these interactions could be mediated by different hydrophobic or aromatic residues within the pocket. For example, the benzene ring of eugenol was stabilized through π-stacking interactions, but these could be mediated by Trp158, Tyr91, or Tyr380, which lie on opposing faces of the binding pocket. In every pose, eugenol also formed extensive hydrophobic interactions with an overlapping complement of aliphatic and aromatic side chains. Moreover,

although eugenol's hydroxyl was consistently oriented towards the only polar amino acid lining the pocket (Ser151), none of the predicted poses adopted a geometry that allowed them to form hydrogen bonds with the surrounding residues. Therefore, recognition of eugenol appears to rely on non-directional hydrophobic interactions formed with a distributed array of binding pocket residues. Although only one of these poses might be energetically favoured, structural studies of odorant binding proteins[21,22] that ferry hydrophobic ligands through the sensory neuron lymph have revealed that an individual odorant can bind in different poses within the same hydrophobic binding cavity; thus, it is possible that eugenol might likewise sample from multiple energetically degenerate binding modes in *Mh*OR5.

To functionally corroborate the eugenol binding site, we identified ten amino acids with side chains that were in close proximity to the ligand density—Val88, Tyr91, Phe92, Ser151, Gly154, Trp158, Met209, Ile213, Tyr380 and Tyr383—and found that mutation of any of these residues to alanine strongly altered eugenol signalling (Fig. 3c, Extended Data Fig. 9a–c). Several of these mutants also displayed increased baseline activity (Extended Data Fig. 9a, e), suggesting that these residues stabilize the closed conformation. Mutation of adjacent residues that project away from the binding site—Thr87 and Leu379—had minimal effect on activation by eugenol (Fig. 3c, Extended Data Figs. 6a, 9a), underscoring the specificity of these perturbations to odorant-dependent gating.

A comparison of the apo and eugenol-bound structures indicates that, in addition to the dilation of the pore, smaller conformational changes appear to be distributed throughout the transmembrane portion of the S0–S6 helices (Extended Data Fig. 10a, b and Supplementary Videos 1, 2). Although the delocalized nature of these small rearrangements makes it challenging to delineate how odorant binding is transduced to pore opening, one potential route is through the S5 helix, which runs parallel to the S7b helix that lines the pore and anti-parallel to the S6 helix that contributes key residues to the odorant-binding pocket. Upon eugenol binding, these three helices move together away from the central axis of the channel, displacing the S7b helices outwards to gate the ion conduction pathway (Extended Data Fig. 10a, b). Close to the extracellular surface of the membrane, the S5 and S7 helices interact through Tyr362 and Leu465, which are highly conserved as hydrophobic amino acids and evolutionarily coupled[23], pointing to a coordinated role in receptor function. These residues remain tightly packed as the S7b helix moves into an open configuration (Extended Data Fig. 10b), suggesting that they might couple conformational rearrangements within the odorant-binding pocket to the dilation of the pore. Mutation of either Tyr362 or Leu465 to alanine impaired eugenol activation, whereas mutation of Tyr362 to phenylalanine had no effect (Extended Data Fig. 10c), supporting a model in which hydrophobic interactions at this position contribute to gating.

## Structural basis of odorant specificity

To investigate the diversity of binding modes used by different ligands, we determined the 2.9 Å structure of *Mh*OR5 in complex with the insect repellent DEET (Extended Data Table 1). The S7b helices in the DEET-bound structure were dilated to a diameter of 8.7 Å (Extended Data Fig. 10d–f), indicating that different ligands elicit a common conformational change to gate the pore. Density corresponding to DEET localized to the same binding pocket as eugenol, encased within the same box-like configuration of aromatic and aliphatic side chains (Fig. 4a, b, Extended Data Figs. 6b, 9d). As with eugenol, computational docking of DEET yielded multiple poses with comparable docking scores that fit the experimental density well (Extended Data Fig. 8a). Whereas each of the top poses was predicted to adopt a distinct orientation, all were stabilized through a similar complement of hydrophobic and/or π-stacking interactions. Although we cannot determine whether DEET adopts only one or multiple conformations within the binding pocket, these observations reinforce how non-directional hydrophobic interactions may contribute to flexible chemical recognition, allowing different

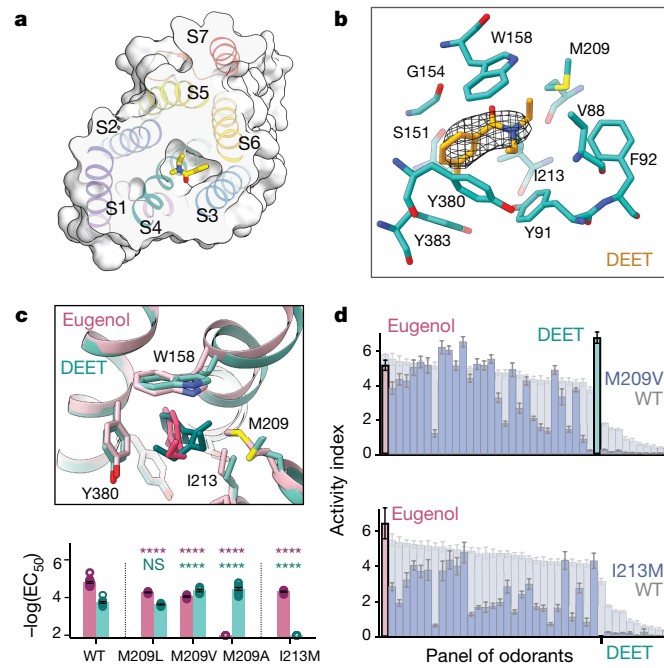

**Fig. 4 | Structure-based mutagenesis retunes *Mh*OR5. a**, Cross-section of the binding pocket of an *Mh*OR5 subunit shown as a surface representation in complex with DEET (shown in stick representation within the pocket). Helices are shown coloured and labelled. **b**, View of the binding pocket (orientation as in Fig. 3b). Black mesh, cryo-EM density for DEET. **c**, Top, overlay of the *Mh*OR5 binding pocket in complex with DEET (teal) and in complex with eugenol (pink). Bottom, effect of mutating Met209 and Ile213 into residues with different length side-chains on eugenol and DEET signalling. Mean ± s.e.m. log(EC$_{50}$) shown. Statistical significance determined using one-way ANOVAs followed by Dunnett's multiple comparison tests comparing mutants to their respective WT controls for each ligand. For mutants for which the EC$_{50}$ was incalculably high and Bartlett's test showed non-homogenous variance, statistical significance determined with a Brown–Forsythe test. ****$P < 0.0001$; NS, not significant. **d**, Tuning curves of *Mh*OR5 M209V (top) and I213M (bottom) mutants in response to a panel of 40 odorants, overlaid and ordered by high-to-low activity index in wild-type *Mh*OR5. Additional dose–response curves shown in Extended Data Fig. 9, additional views of the pocket in Extended Data Figs. 6, 9d, and *n* values for all biological replicates in Supplementary Tables 2, 4, 5.

ligands to bind to the same structural locus or potentially enabling a single odorant to sample from multiple poses within the binding cavity.

To investigate whether the broader panel of *Mh*OR5 ligands is recognized through a similar structural logic, we examined how their physicochemical descriptors correlated with receptor activity. Multiple regression analysis revealed that although no single metric was strongly predictive of agonism, the descriptors that best accounted for *Mh*OR5 activity were low polar surface area, low water solubility, and low potential for forming hydrogen bonds (Extended Data Table 2), consistent with our structural observations of a geometrically simple binding site in which diffuse hydrophobic interactions dominate. *Mh*OR1 agonism was less correlated with these descriptors, suggesting that they have a heterogeneous role in shaping the tuning of different receptors (Extended Data Table 2). Furthermore, the top 31 *Mh*OR5 agonists identified in our panel could be docked within this same binding site with favourable scores, stabilized predominantly through hydrophobic interactions (Extended Data Fig. 8), suggesting that diverse odorants can be recognized through distributed and non-directional interactions with an overlapping subset of residues in the *Mh*OR5 binding pocket.

A comparison of the eugenol and DEET-bound structures reveals how the *Mh*OR5 binding pocket might accommodate such diverse ligands. The constellation of amino acids lining the binding pocket retains the same overall geometry in both structures, leaving the architecture of the hydrophobic box largely unchanged. However, a small displacement of the S4 helix results in expansion of the pocket, probably to accommodate the longer aliphatic moiety of DEET and avoid a steric clash with the side chain of Met209 (Fig. 4c, Extended Data Fig. 6b). Functional data support these structural observations. Mutation of Met209 to smaller hydrophobic amino acids (valine or alanine) enhanced the affinity of DEET (Fig. 4c, Extended Data Fig. 9b). The same mutations attenuated eugenol sensitivity, suggesting that this smaller odorant occupies the binding pocket less optimally in the absence of the bulky methionine side chain. Conversely, mutation of Ile213, another aliphatic S4 residue that lies in close proximity to DEET, to the larger residue methionine abolished DEET sensitivity but marginally altered eugenol signalling (Fig. 4c, Extended Data Fig. 9c). Structure-guided mutagenesis therefore differentially altered the sensitivity of *Mh*OR5 to these two ligands. Furthermore, the I213M and M209V mutations broadly reconfigured the tuning of *Mh*OR5 to a larger panel of 40 odorants (Fig. 4d), supporting a model in which diverse chemicals are recognized by shared structural elements within a common binding pocket. Changes in odorant tuning, however, did not adhere to a simple logic, consistent with the complexity of physicochemical properties that define *Mh*OR5 agonism (Extended Data Table 2) and with the proposal that both the global geometry and local chemical environment of the binding pocket contribute to its chemical sensitivity.

To assess whether *Mh*OR5 can serve as a structural model for chemical recognition in other ORs, we used sequence homology to identify ten residues predicted to line the binding pocket in the more narrowly tuned *Mh*OR1 and examined their contribution to odorant tuning (Extended Data Fig. 11a–c). For all but one of these residues, mutation to alanine impaired *Mh*OR1 activation by its ligands, 1-octanol and eugenol, indicating that the odorant binding pocket is a conserved structural feature of this family, even between divergent receptors that display distinct chemical tuning. Furthermore, mutation of Met231 in *Mh*OR1 to the corresponding residue in *Mh*OR5, isoleucine, enhanced the sensitivity of *Mh*OR1 to a panel of odorants (Extended Data Fig. 11d). Thus, whereas the I213M mutation narrows the chemical tuning of *Mh*OR5, the reciprocal M231I mutation broadens the molecular receptive range of *Mh*OR1, shifts in sensitivity that could be attributed to alterations in the size of the binding pocket. Odorant recognition in different insect olfactory receptors appears therefore to rely on a conserved binding site that can be readily retuned to detect different regions of chemical space.

## Discussion

The broad tuning of olfactory receptors is central to the detection and discrimination of the vast chemical world. Here we show that *Mh*OR5 detects a wide array of odorants through a single promiscuous binding site, offering structural insight into how such flexible chemical recognition is achieved. Notably, odorant binding relies predominantly on hydrophobic interactions, which lack the strict geometric constraints inherent to other intermolecular associations (such as hydrogen bonds) that frequently mediate ligand recognition. The distributed arrangement of hydrophobic and aromatic residues across multiple surfaces of the binding pocket further relaxes orientational constraints by allowing odorants to form comparable interactions with many of its faces. Moreover, the simple geometry of the binding site imposes minimal restriction on the shape of odorants that can bind, accommodating both eugenol and DEET with little structural rearrangement. Computational docking analyses support these structural observations and suggest that the same logic underlies the sensitivity of *Mh*OR5 to structurally and chemically diverse ligands. The prevalence of comparatively weak intermolecular interactions is compatible with the low affinity of most odorants[11,13,17,24] and the small conformational change required to gate the channel. Olfactory receptor tuning thus depends on the stereochemistry of its ligands[25,26], but does not adhere to the classic lock-and-key mechanism that governs many receptor–ligand interactions.

Residues that have been implicated in odorant specificity in different neopteran receptors map to the binding pocket of *Mh*OR5[10,27–30], indicating that it represents a conserved and canonical locus for odorant detection across this highly divergent family. Binding of DEET to the same site offers structural corroboration that this insect repellent might exploit the promiscuity of diverse ORs and serve as a molecular 'confusant' by scrambling the olfactory code[31]. Other modulators of olfactory receptors, such as VUAA1 (which inhibits *Mh*OR5), cannot favourably dock within this binding pocket owing to their much larger size, suggesting that insect olfactory receptors might possess additional points of allosteric modulation that expand their signalling mechanisms.

Several important implications arise from our observation that diverse odorants share the same structural determinants for binding. Notably, even single conservative mutations within the binding pocket can broadly reconfigure the chemical tuning of the receptor, a feature that is likely to have facilitated the rapid evolution of receptors with distinct ligand specificity[2,27–29]. However, such extensive retuning also poses a substantial evolutionary constraint, as individual binding-site mutations are likely to have a pleiotropic effect on the representation of multiple odorants, potentially serving to broadly reconfigure the odour code. The promiscuous and arbitrary nature of odorant recognition is likely to impose substantial selective pressures on the structure and function of olfactory circuits, driving the evolution of synaptic and circuit mechanisms that can decorrelate, decode, and impose meaning onto combinatorial patterns of receptor activity[32]. Odour discrimination is thus transformed from a biochemical problem at the receptor level to a neural coding problem within the brain.

Although the structure of a mammalian olfactory receptor has yet to be elucidated, odorant detection in mammals has been proposed to also rely on distributed hydrophobic and non-directional interactions within a deep transmembrane pocket[33–35]. Structurally and mechanistically distinct receptor families appear to therefore rely on similar principles for their broad chemical tuning, pointing to common constraints in how diverse hydrophobic molecules can be recognized. Additional mechanisms for odorant recognition certainly exist, in particular for receptors that are selectively tuned to ethologically relevant chemical classes, such as pheromones[36], the perceptual meaning of which is singular and invariant. Whether stricter odorant specificity relies on distinct intermolecular binding modes, variations in the geometry of the binding pocket, or both, remains to be determined.

Finally, our work sheds light on the evolution of the insect olfactory system. We demonstrate that *Mh*ORs can function as homomeric odorant-gated channels, supporting the proposal that they lie at the ancestral origin of the insect olfactory receptor family[4,14], which expanded massively across insect lineages to emerge as possibly the largest and most divergent class of ion channels in nature[2]. Why neopteran ORs became obligate heteromers with Orco remains unclear, but presumably reflects the fact that Orco confers structural stability on the complex, thereby relaxing evolutionary constraints on the ORs and allowing them to further diversify, to ultimately support the flexible detection and discrimination of an enormous and ever-changing chemical world.

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

## Methods

### Expression and purification of *Mh*OR5

The coding sequence of *M. hrabei* OR5 (*Mh*OR5) was synthesized as a gene fragment (Twist). Residues Lys2 to Pro474 were cloned into a pEG BacMam vector[37] containing N-terminal tags of Strep II, super-folder GFP[38], and an HRV 3C protease site for cleavage (N-CACC<u>atg</u>-ST 2-SGR-sfGFP-PPX-AscI-*Mh*OR5-taa-NotI-C). The AscI/NotI restriction enzyme sites enable efficient cloning of different OR sequences. SF9 cells (ATCC CRL-1711) were used to produce baculovirus containing the *Mh*OR5 construct, and the virus, after three rounds of amplification, was used to infect HEK293S GnTI⁻ cells (ATCC CRL-3022)[37]. Cell lines were not authenticated except as performed by the vendor. HEK293S GnTI⁻ cells were grown at 37 °C with 8% carbon dioxide in Freestyle 293 medium (Gibco) supplemented with 2% (v/v) fetal bovine serum (Gibco). Cells were grown to $3 \times 10^6$ cells/ml and infected at a multiplicity of infection of about 1. After 8–12 h, 10 mM sodium butyrate (Sigma-Aldrich) was added to the cells and the temperature was dropped from 37 °C to 30 °C for the remainder of the incubation. Seventy-two hours after initial infection, cells were collected by centrifugation, washed with phosphate-buffered saline (pH 7.5; Gibco), weighed and flash frozen in liquid nitrogen. Pellets were stored at −80 °C until they were thawed for purification.

For purification, cell pellets were thawed on ice and resuspended in 20 ml lysis buffer per gram of cells. Lysis buffer was composed of 50 mM HEPES/NaOH (pH 7.5), 375 mM NaCl, 1 μg/ml leupeptin, 1 μg/ml aprotinin, 1 μg/ml pepstatin A, 1 mM phenylmethylsulfonyl fluoride (PMSF; all from Sigma-Aldrich) and about 3 mg DNase I (Roche). *Mh*OR5 was extracted using 0.5% (w/v) *n*-dodecyl β-D-maltoside (DDM; Anatrace) with 0.1% (w/v) cholesterol hemisuccinate (CHS; Sigma-Aldrich) for 2 h at 4 °C. The mixture was clarified by centrifugation at 90,000*g* and the supernatant was added to 0.1 ml StrepTactin Sepharose resin (GE Healthcare) per gram of cells and rotated at 4 °C for 2 h. The resin was collected and washed with 10 column volumes (CV) of 20 mM HEPES/NaOH, 150 mM NaCl with 0.025% (w/v) DDM and 0.005% (w/v) CHS (together, SEC buffer). *Mh*OR5 was eluted by adding 2.5 mM desthiobiotin (DTB) and cleaved overnight at 4 °C with HRV 3C Protease (EMD Millipore). Sample was concentrated to about 5 mg/ml and injected onto a Superose 6 Increase column (GE Healthcare) equilibrated in SEC buffer. Peak fractions containing *Mh*OR5 were concentrated until the absorbance at 280 nm reached 5–6 (approximately 5 mg/ml) and immediately used for grid preparation and data acquisition. For the eugenol-bound structure, peak fractions were pooled, and eugenol (Sigma Aldrich, CAS#97-53-0) dissolved in dimethylsulfoxide (DMSO; both Sigma-Aldrich) was added for a final odour concentration of 0.5 mM, and the complex was incubated at 4 °C for 1 h. The maximum DMSO concentration was kept below 0.07%. The complex was then concentrated to approximately 5 mg/ml and used for grid preparation. For the DEET-bound structure, sample from the overnight cleavage step was concentrated to about 5 mg/ml and injected into the Superose 6 Increase column equilibrated in SEC buffer with 1 mM DEET (Sigma Aldrich, CAS#134-62-3). Peak fractions were concentrated to about 5 mg/ml and used immediately for grid preparation.

### Cryo-EM sample preparation and data acquisition

Cryo-EM grids were frozen using a Vitrobot Mark IV (FEI) as follows: 3 μl of the concentrated sample was applied to a glow-discharged Quantifoil R1.2/1.3 holey carbon 400 mesh gold grid, blotted for 3–4 s in >90% humidity at room temperature, and plunge frozen in liquid ethane cooled by liquid nitrogen.

Cryo-EM data were recorded on a Titan Krios (FEI) operated at 300 kV, equipped with a Gatan K2 Summit camera. SerialEM[39] was used for automated data collection. Movies were collected at a nominal magnification of 29,000× in super-resolution mode resulting in a calibrated pixel size of 0.51 Å/pixel, with a defocus range of approximately −1.0 to −3.0 μm. Fifty frames were recorded over 10 s of exposure at a dose rate of 1.22 electrons per Å² per frame.

Movie frames were aligned and binned over $2 \times 2$ pixels using Motion-Cor2[40] implemented in Relion 3.0[41], and the contrast transfer function parameters for each motion-corrected image were estimated using CTFFIND4[42].

**Apo structure.** Two datasets were collected with 4,050 micrographs in dataset A and 3,748 micrographs in dataset B. Processing was done independently for each dataset in the following way: particles were picked using a 3D template generated in an initial model from a dataset of 5,000 particles picked in manual mode. A total of 562,794 (dataset A) and 536,145 (dataset B) particles were subjected to 2D classification using RELION-3.0[41]. Particles from the best 2D classes (210,833 for dataset A, 183,061 for dataset B) were selected and subjected to 3D classification imposing C4 symmetry and adding a soft mask to exclude the detergent micelle after 25 iterations. One class from each dataset containing 44,884 (dataset A) and 43,788 (dataset B) particles was clearly superior in completeness and definition of the transmembrane domains. These particles were subjected to 3D refinement with C4 symmetry, followed by Bayesian polishing and CTF refinement. The polished particles from both datasets were exported to cryoSPARC v2[43] and processing continued with the joined dataset of 88,672 particles. In cryoSPARC, further heterogeneous refinement resulted in a single class with 49,832 particles that were subjected to particle subtraction with a micelle mask. Non-uniformed refinement of subtracted particles imposing C4 symmetry yielded the final map with an overall resolution of 3.3 Å as estimated by cryoSPARC with a cutoff for the Fourier shell correlation (FSC) of 0.143[44].

**Ligand-bound structures.** Processing for the eugenol-bound and DEET-bound structures occurred through the following pipeline: 4,410 (eugenol) and 4,365 (DEET) micrographs were collected and used to pick 461,254 (eugenol) and 787,448 (DEET) particles that were extracted, unbinned and exported into cryoSPARC v2. In cryoSPARC, several rounds of 2D classification resulted in 221,339 (eugenol) and 180,874 (DEET) particles that were used to generate an initial model with four classes with no imposed symmetry. These models were inputted as templates of a heterogeneous refinement with no imposed symmetry, from which one (eugenol) and two (DEET) final classes were selected containing 129,031 (eugenol) and 121,441 (DEET) particles. These particles were refined and exported to RELION 3.0 where they were subjected to a round of 3D classification with no imposed symmetry. The best class from this 3D classification contained 54,900 (eugenol) and 56,191 (DEET) particles that were subjected to Bayesian polishing and CTF refinement. Polished particles were then imported into cryoSPARC v2 and subjected to particle subtraction. Final non-uniform refinement with C4 symmetry imposed resulted in the final maps with overall resolution of 2.9 Å in both cases, estimated with a cutoff for the FSC of 0.143. In all cases, the four-fold symmetry of the channel was evident from the initial 2D classes without having imposed symmetry and refinements without imposed symmetry produced four-fold symmetric maps.

### Model building

The Cryo-EM structure of Orco (Protein Data Bank (PDB) accession 6C70) was used as a template for homology modelling of *Mh*OR5 using Modeller[45], followed by manual building in Coot[46]. The 3.3 Å density map of the apo was of sufficient quality to build the majority of the protein, with the exception of the S3–S4 and S4–S5 loops, the 13 N-terminal residues and the 5 C-terminal residues. The models were refined using real-space refinement implemented in PHENIX[47] for five macro-cycles with four-fold non-crystallographic symmetry applied and secondary structure restraints applied. The eugenol- and DEET-bound models were refined including the ligands, which were placed as a starting point within the corresponding density in a pose that was obtained through

docking methods (described below) and with restraints obtained with the electronic Ligand Builder and Optimization Workbench[58] (eLBOW) implemented in PHENIX. Model statistics were obtained using Mol-Probity. Models were validated by randomly displacing the atoms in the original model by 0.5 Å, and refining the resulting model against half maps and full map[48]. Model–map correlations were determined using phenix.mtriage. Images of the maps were created using UCSF ChimeraX[49]. Images of the model were created using PyMOL[50] and UCSF ChimeraX[49].

## Docking analysis

All compounds were docked using Glide[20,51] implemented in Maestro (Schrödinger, suite 2020). In brief, the model was imported into Maestro and prepared for docking. A 20 Å³ cubical grid search was built centred in the region of observed ligand density. Ligand structures were imported into Maestro by their SMILES unique identifiers and prepared using Epik[52] to generate their possible tautomeric and ionization states, all optimized at pH 7.0 ± 2. All ligands were docked within the built grid, and the top poses that best fit the density are presented in Extended Data Fig. 8. The top activators scored with values between −7.4 and −4. While all activators docked with negative scores, some non-activators also docked with favourable scores. For example, caffeine docked favourably despite the molecule not activating the channel in our functional experiments. As docking does not incorporate dynamics of the receptor, it is not expected that docking will correlate homogeneously or monotonically with experimentally determined activity of ligands. At most a qualitative agreement can be expected.

## Structure analysis

Residues at subunit interfaces were identified using PyMOL as any residue within 5 Å of a neighbouring subunit (Extended Data Fig. 5d). The pore diameters along the central axis and lateral conduits were calculated using the program HOLE[53], which models atoms as solid spheres of Van der Waals radius (Fig. 2a–c, Extended Data Fig. 10d, e). Two calculations were performed for each structure: one along the central four-fold axis (central pore) and another between subunits near the cytosolic membrane interface (lateral conduits). The plots in Fig. 2b and Extended Data Fig. 10e show the diameter along the central axis of the main conduit and the lateral conduit. The measurements in Fig. 2d and Extended Data Fig. 10f between residues lining the pore are taken from atom centres using PyMol. Electrostatic surface representations were performed using ChimeraX v1.1, coulombic estimation with default parameters (Extended Data Fig. 7). Morph videos were created in ChimeraX v1.1 with direct interpolation between states.

## Electrophysiology

HEK293 cells were maintained in high-glucose Dulbecco's modified Eagle's medium (DMEM) supplemented with 10% (v/v) fetal bovine serum (FBS) and 1% (v/v) GlutaMAX (all Gibco) at 37 °C with 5% (v/v) carbon dioxide. Cells were plated on 35-mm tissue-culture-treated Petri dishes 72–48 h before recording, and infected with the same pEG BacMam GFP-tagged *Mh*OR5 construct used for expression 24–48 h before recording. Electrodes were drawn from borosilicate patch glass (Sutter Instruments) and polished (MF-83, Narishige Co.) to a resistance of 3–6 MΩ when filled with pipette solution. Analogue signals were digitized at 20 kHz (Digidata 1440A, Molecular Devices) and filtered at 1 kHz (whole-cell) or 2 kHz (patch recordings) using the built-in four-pole Bessel filter of a Multiclamp 700B patch-clamp amplifier (Molecular Devices) in whole-cell or patch mode. Whole-cell recordings were baseline-subtracted offline. Patch signals were further resampled offline for representations.

Whole-cell and single-channel recordings in Fig. 1c and Extended Data Fig. 2e were performed using an extracellular (bath) solution composed of 135 mM NaCl, 5 mM KCl, 2 mM MgCl₂, 2 mM CaCl₂, 10 mM glucose, 10 mM HEPES-Na/HCl (pH 7.3, 310 mOsm/kg) and an intracellular (pipette)

solution composed of 150 mM KCl, 10 mM NaCl, 1 mM EDTA-Na, 10 mM HEPES-Na/HCl (pH 7.45, 310 mOsm/kg). Single-channel recordings were done in excised outside-out mode. Stock eugenol solution was prepared by dissolving in DMSO at 150 mM, and working solutions were prepared by diluting stocks to 3 μM in extracellular solution. Solutions were locally perfused using a microperfusion system (ALA Scientific Instruments).

## Cell-based GCaMP fluorescence calcium flux assay

All DNA constructs used in this assay were cloned into a modified pME18s vector with no fluorescent marker, flanked by AscI/NotI restriction enzyme sites for efficient cloning. Each transfection condition contained 0.5 μg of a plasmid encoding GCaMP6s (Addgene #40753) and 1.5 μg of the plasmid encoding the appropriate olfactory receptor, diluted in 250 μl OptiMEM (Gibco). In experiments with heteromeric olfactory receptors, the total amount of DNA was 1.5 μg, in a ratio of 1:1 of Orco:OR. These were diluted in a solution containing 7 μl Lipofectamine 2000 (Invitrogen) and 250 μl OptiMem, followed by a 20-min incubation at room temperature. HEK293 cells were maintained in high-glucose DMEM supplemented with 10% (v/v) FBS and 1% (v/v) GlutaMAX at 37 °C with 5% (v/v) carbon dioxide. Cells were detached using trypsin and resuspended to a final concentration of 1 × 10⁶ cells/ml. Cells were added to each transfection condition, mixed and added to 2 × 16 wells in a 384-well plate (Grenier CELLSTAR). Four to six hours later, a 16-port vacuum manifold on low vacuum was used to remove the transfection medium, replaced by fresh FluoroBrite DMEM (Gibco) supplemented with 10% (v/v) FBS and 1% (v/v) GlutaMAX. Twenty-four hours later, this medium was replaced with 20 μl reading buffer (20 mM HEPES/NaOH (pH 7.4), 1× HBSS (Gibco), 3 mM Na₂CO₃, 1 mM MgSO₄, and 2 or 5 mM CaCl₂) in each well. The calcium concentration was optimized for each receptor to account for their differences in baseline activity: for experiments with *Mh*OR5 and *Mh*OR5 mutants, reading buffer contained 2 mM CaCl₂, while 5 mM CaCl₂ was used for *Mh*OR1, Orco and Orco–*Ag*OR28 heteromers. The fluorescence emission at 527 nm, with excitation at 480 nm, was continuously read by a Hamamatsu FDSS plate reader. After 30 s of baseline recording, an optimized amount of odorant solution−10 μl for all *Mh*OR-containing experiments or 20 μl for all Orco-containing experiments−was added to the cells and read for 2 min. All solutions were warmed to 37 °C before beginning.

Seven ligand concentrations were used for each transfection condition in sequential dilutions of 3, alongside a control well of only reading buffer. Ligands were dissolved in DMSO to 150 mM, then diluted with reading buffer to a highest final-well concentration of 0.5 mM (DMSO never exceeded 0.5%). Water-soluble ligands (arabinose, caffeine, denatonium, glucose, MSG, sucrose) were dissolved directly into reading buffer. If experimental data indicated a more sensitive response than this range, the concentration was adjusted accordingly. Ligand concentrations for mutants were the same as for the corresponding wild-type OR. Each plate contained a negative control of GCaMP6s transfected alone and exposed to eugenol for *Mh*OR5 and VUAA1 for Orco experiments. Additionally, each plate included the corresponding wild-type OR with its cognate ligand−*Mh*OR5 and *Mh*OR1 with eugenol, Orco with VUAA1, and Orco–*Ag*OR28 with acetophenone−as a positive control to account for plate-to-plate variation in transfection efficiency and cell count. A control of DMSO alone was also tested to ensure no activity effects were due to the solvent. Each concentration of ligand was applied to four technical replicates, which were averaged and considered a single biological replicate.

The baseline fluorescence (*F*) was calculated as the average fluorescence of the 30 s before odour was added to the plate. Within each well, Δ*F* was calculated as the difference between the average of the last 10 s of fluorescence and the baseline *F*. Δ*F*/*F* was then calculated as the Δ*F* divided by the baseline fluorescence (*F*). Finally, the Δ*F*/*F* for each concentration was normalized to the maximum Δ*F*/*F* value of the corresponding positive control present on each plate: *Mh*OR5 and *Mh*OR1

with eugenol, Orco with VUAA1, and Orco–*Ag*OR28 with acetophenone to account for inevitable variations in transfection efficiency and cell counts across different plates. The normalized $\Delta F/F$ averaged across all experiments for a given condition is the value used to construct the dose–response curves in all plots (Figs. 1b, 2e–g, Extended Data Figs. 2d, 9a–c, 10c, 11b). All wild-type curves come from the same plates as the experimental data in the same plot. Baseline values for wild-type and mutant channels were found by normalizing each $F$ value by the negative GCaMP6s-only control on the same plate (Extended Data Figs. 1c, 9a, e).

For all experiments, GraphPad Prism 8 was used to fit the dose–responses curves to the Hill equation from which the $EC_{50}$ of the curve was extracted. Three metrics were used to characterize the dose–response curve for each ligand: activity index, $\log(EC_{50})$ and max $\Delta F/F$. For conditions where $EC_{50}$ was too high for the dose–response curve to reach saturation and therefore could not be fitted to a Hill equation, a value of $-2$ was assigned to the $EC_{50}$, which is more than an order of magnitude higher than the highest concentration used. Max $\Delta F/F$ is the maximum response achieved at the highest concentration. Activity index is defined as the negative product of $\log(EC_{50})$ and max $\Delta F/F$, as follows:

$$\text{Activity index} = -\log(EC_{50}) \times \max \Delta F/F$$

## Gels and small-scale transfections

For western blots and fluorescence-detection size-exclusion chromatography (FSEC) traces (Extended Data Figs. 1a, b, 9g), HEK293 cells were maintained in high-glucose DMEM supplemented with 10% (v/v) FBS and 1% (v/v) GlutaMAX at 37 °C with 5% (v/v) carbon dioxide. Cells were detached using trypsin and plated in six-well plates at a concentration of $0.4 \times 10^6$ per well. Twenty-four hours later, cells were transfected with 2 μg of DNA in the same superfolder GFP-containing pEG BacMam vector used for large-scale purification and 9 μl Lipofectamine 2000 (Invitrogen) diluted in 700 μl OptiMEM and added dropwise to the cells after a 5-min incubation. Twenty-four hours later, cells were checked for GFP fluorescence, rinsed with phosphate-buffered saline, and collected by centrifugation. Cells were either frozen at −20 °C or used immediately.

Cell pellets were rapidly thawed and resuspended in 200 μl lysis buffer containing 50 mM HEPES/NaOH (pH 7.5), 375 mM NaCl, an EDTA-free protease inhibitor cocktail (Roche), and 1 mM PMSF. The protein was extracted for 2 h at 4 °C by adding 0.5% (w/v) DDM with 0.1% (w/v) CHS after 10 s sonication in a water bath. This mixture was then clarified by centrifugation and filtered. The supernatant was added to a Shimadzu autosampler connected to a Superose 6 Increase column equilibrated in SEC buffer. An aliquot of the supernatant was also used to run SDS–PAGE (Bio-Rad, 12% Mini-PROTEAN TGX) and Blue Native(BN)-PAGE (Invitrogen, 3–12% Bis-Tris) gels. Gels were transferred using Trans-Blot Turbo Transfer Pack (Bio-Rad) and blocked overnight. The following day, gels were stained with rabbit anti-GFP polyclonal antibody (Life Technologies; 1:20,000), washed, incubated with anti-rabbit secondary antibody (1:10,000), and imaged with ImageLab.

## Lifetime sparseness calculation

The lifetime sparseness[54,55] measure in Extended Data Fig. 1d was used to quantify olfactory receptor tuning breadth and calculated as follows:

$$\text{Lifetime sparseness} = \left(\frac{1}{1-\frac{1}{n}}\right) \times \left(1 - \frac{\left(\sum_{i=1}^n \frac{\text{res}_i}{n}\right)^2}{\sum_{i=1}^n \frac{\text{res}_i^2}{n}}\right),$$

in which $n$ is the number of ligands in the set, and $\text{res}_i$ is the receptor's response to a given ligand $i$. All inhibitory responses (values below 0) were set to 0 before the calculation[54,55]. The *Drosophila melanogaster* OR dataset comes from the DoOR database[56].

## Multiple regression analysis

A set of 11 molecular descriptors were compiled for all 54 ligands tested from PubChem, Sigma-Aldrich, ChemSpider, EPA, and The Good Scents Company; the values used are in Supplementary Table 9. A multiple regression analysis using the scikit-learn Linear Regression module was used to assess the accuracy with which the receptor activity could be predicted by individual descriptors (1-dimensional analysis) or combinations of two descriptors (2-dimensional analysis) (Extended Data Table 2). Owing to the absence of reported metrics for some ligands—acetic acid, citric acid, MSG, sucrose, denatonium, and VUAA1—the analysis was performed on the remaining 48 ligands. For the 1-dimensional analysis, a single variable linear regression was performed for each descriptor independently. The analysis sought to fit a linear model with coefficients $w1, …, wn+1$, in which $n$ is the dimension of the input data. The optimal coefficient set was determined using residual sum of squares optimization between the observed activity index targets and those predicted by linear approximation using solved coefficients. This process was repeated for the 2-dimensional case, using every unique permutation of descriptors across the 11-dimensional space. As a means of assessing the predictive power of a given combination, the $R^2$-value, reflecting the square of the correlation coefficient between observed and modelled values of the activity index, was calculated for each linear model and reported in Extended Data Table 2. This allowed ranking of descriptor sets based on accuracy of prediction.

## Sequence alignments

For Extended Data Fig. 11a, the alignment between the sequences of *Mh*OR1 and *Mh*OR5 was done using MAFFT implemented in JalView[57] with minimal manual adjustment based on the structure of *Mh*OR5. For Extended Data Fig. 5a, the sequence alignment between *A. bakeri* Orco and *Mh*OR5 was done by aligning the published structure of *A. bakeri* Orco (PDB 6C70) and the structure of *Mh*OR5 in PyMOL. All sequence alignments were visualized and plotted using JalView[57].

## Reporting summary

Further information on research design is available in the Nature Research Reporting Summary linked to this paper.

## Data availability

All data underlying this study are available upon request from the corresponding author.

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

**Acknowledgements** We thank R. Axel, R. MacKinnon, J. Chen, S. Klinge, B. Noro, J. Butterwick, M. Maldonado, C. McBride and members of the Ruta laboratory for discussion and comments on the manuscript, R. Hite and T. Walz for advice on cryo-EM processing; P. Stock for advice on the multiple regression analysis; P. Brand for advice on receptor alignments; L. Ramos-Espiritu for support on functional assays; and M. Ebrahim and J. Sotiris at The Rockefeller University Evelyn Gruss Lipper Cryo-Electron Microscopy Resource Center for assistance with microscope operation. This work was supported by a Leon Levy Postdoctoral Fellowship (to J.d.M.) and the National Institutes of Health (K99DC019401 to J.d.M. and R01AI103171 to V.R.).

**Author contributions** V.R. and J.d.M. conceived the study, designed experiments and wrote the manuscript, with input from M.A.Y. J.d.M. purified *Mh*OR5, performed electrophysiology, collected and analysed cryo-EM data, built and refined the models, and performed molecular docking. M.A.Y. performed molecular biology, cell culture and calcium imaging assays.

**Competing interests** The authors declare no competing interests.

**Additional information**
**Correspondence and requests for materials** should be addressed to V.R.

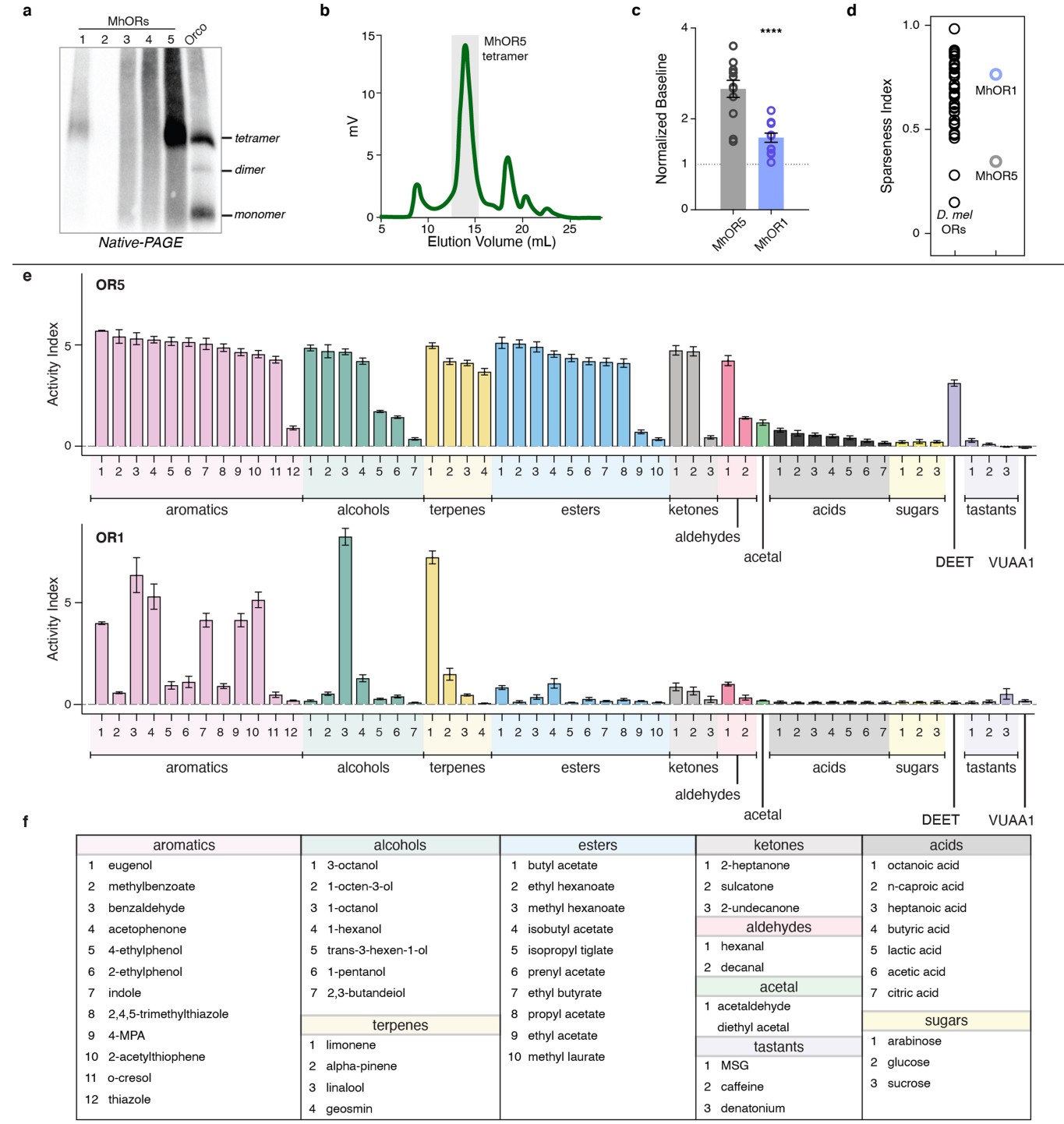

**Extended Data Fig. 1 | Biochemical and functional comparison of *Mh*OR5 and *Mh*OR1. a**, Representative image of western blots of all five *Mh*ORs and *A. bakeri* Orco fused to GFP in non-denaturing Blue Native gels, stained with anti-GFP antibodies. Position of the Orco tetramer, dimer and monomer marked to indicate stoichiometry. Full lanes are shown. This experiment was repeated five times with comparable results. **b**, Size-exclusion chromatography (SEC) trace of *Mh*OR5. **c**, Baseline fluorescence of *Mh*OR5 and *Mh*OR1 in the functional calcium assay, normalized to a GCaMP-only control on the same plate (*n* = 12). Statistical significance was determined using an unpaired, two-tailed *t*-test. ****$P$ < 0.0001. **d**, Comparison of breadth of tuning, measured as lifetime sparseness (see Methods), of *D. melanogaster* ORs and *Mh*ORs. Lifetime sparseness close to 0 suggests broad tuning whereas 1 indicates selective tuning to a single ligand in the panel. **e**, **f**, Receptor activity for *Mh*OR5 (**e**, top) and *Mh*OR1 (**e**, bottom) with ligands (named in **f**) sorted by chemical classes. *n* values for biological replicates provided in Supplementary Tables 1, 6, 7. Data are presented as mean values ± s.e.m.

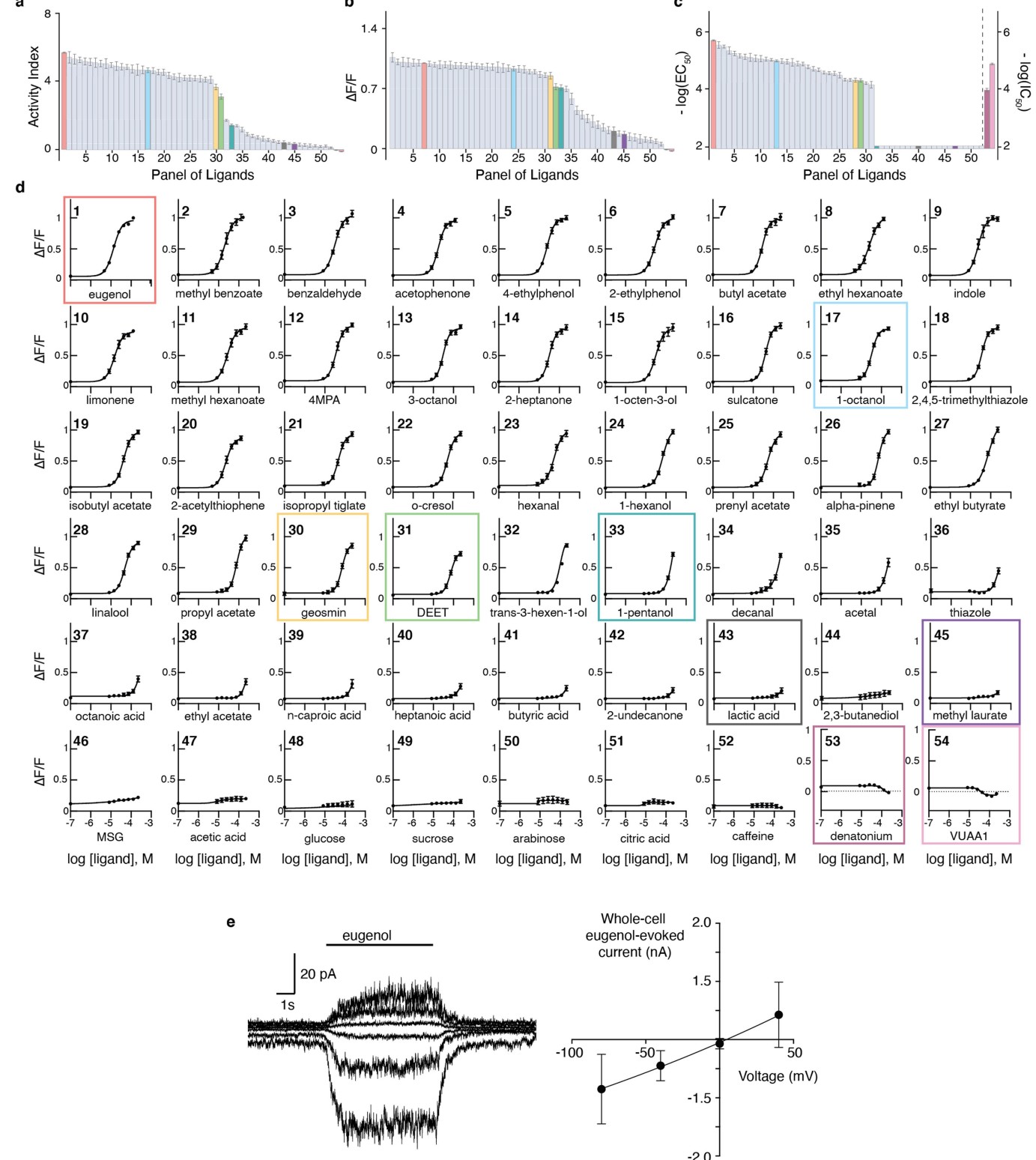

**Extended Data Fig. 2 | *Mh*OR5 is activated by a broad set of odorants.**
**a**–**c**, Tuning curves of *Mh*OR5 sorted by activity index (**a**), max Δ*F/F* (**b**), and −log(EC₅₀) (**c**). Activity index was calculated as −log(EC₅₀) × max Δ*F/F*. In **c**, the inhibitory ligands VUAA1 and denatonium are shown as IC₅₀. More details about these measurements in Methods. **d**, Dose–response curves for all the individual ligands shown in **a**–**c**, averaged across all experiments and shown with s.e.m. Note that the activity index, which combines −log(EC₅₀) and max Δ*F/F*, captures agonism by ligands with low affinity such as 1-pentanol (teal) or

sub-maximal efficacy such as DEET (green). Relevant summary tables, including *n* values for biological replicates, are in Supplementary Tables 1, 8. **e**, Representative traces of an excised outside-out patch of HEK cells expressing *Mh*OR5 and stimulated with 3 µM eugenol at voltages from −60 mV to 40 mV. Right, current–voltage (*IV*) plot of eugenol-elicited currents from whole-cell recordings of HEK cells expressing *Mh*OR5 (shown as absolute current measurements without normalization, with s.d. from *n* independent replicates; *n*₋₈₀ₘᵥ = 8, *n*₋₄₀ₘᵥ = 7, *N*₀ₘᵥ = 7, *n*₄₀ₘᵥ = 4).

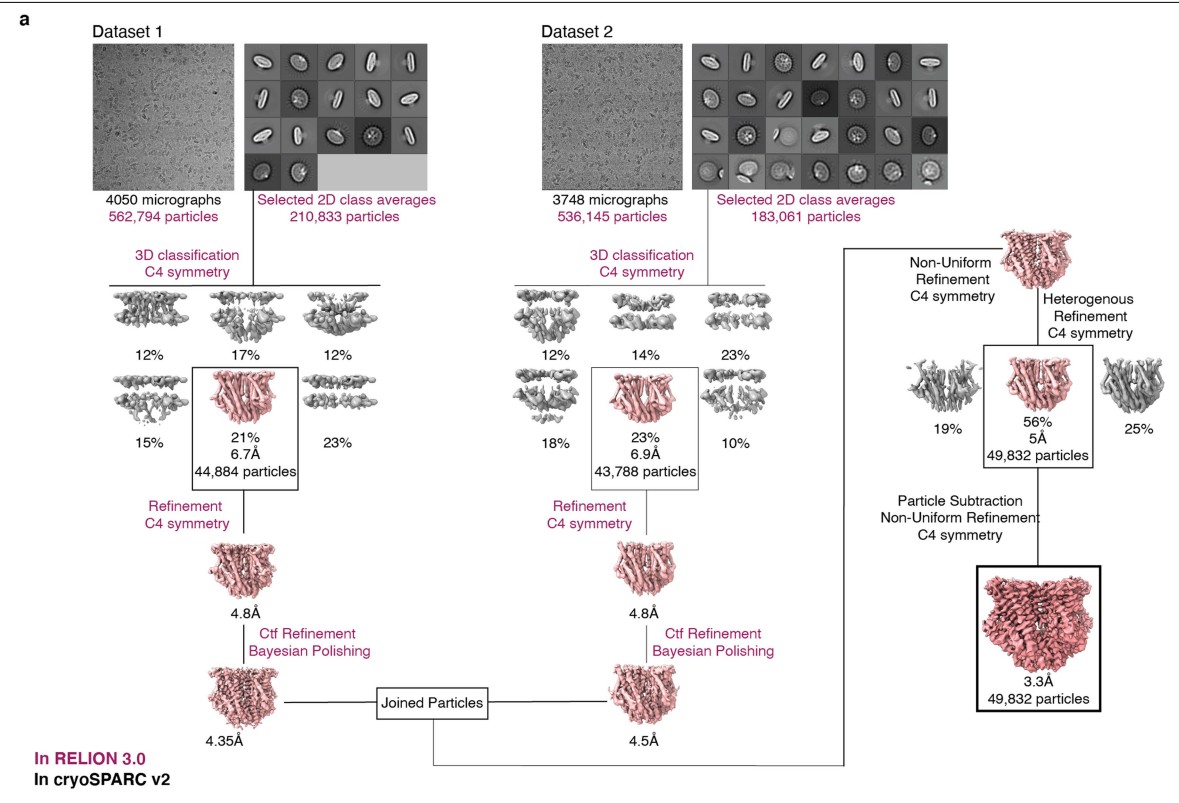

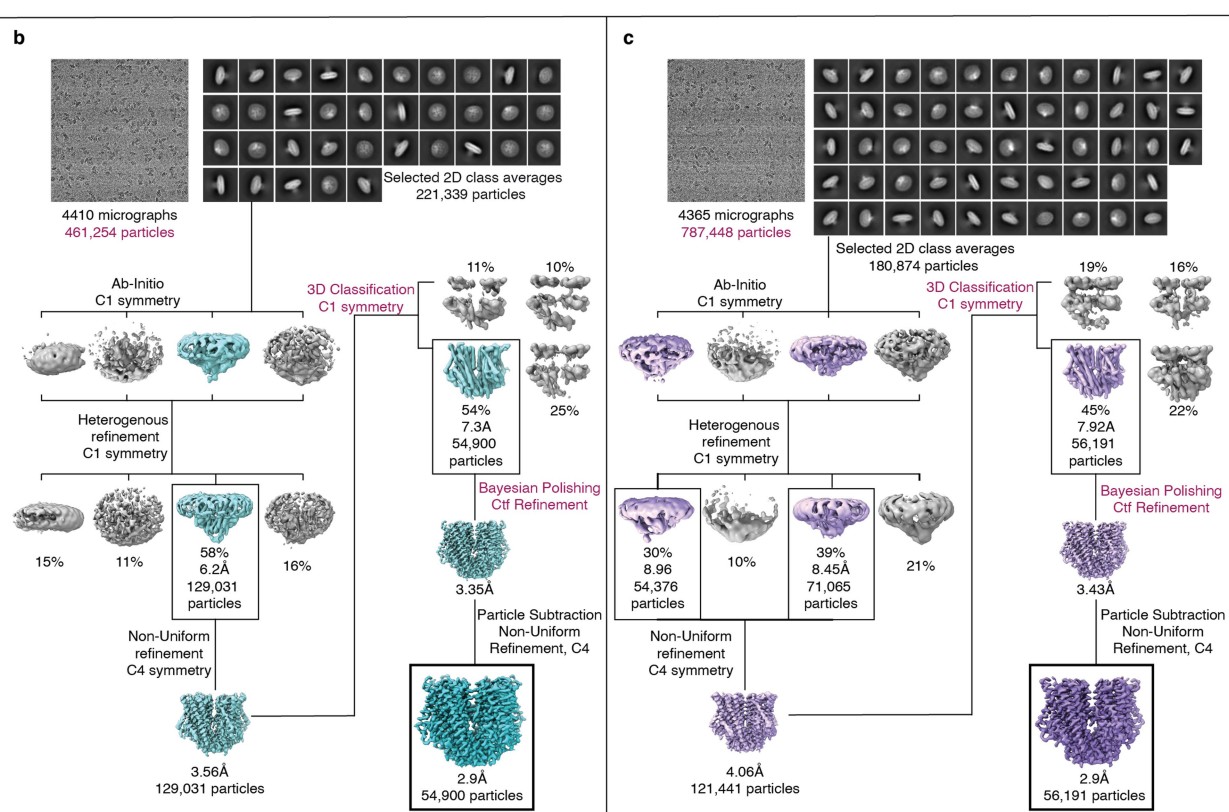

**Extended Data Fig. 3 | Cryo-EM data analysis. a–c,** Processing pipelines for the apo structure (**a**), the eugenol-bound structure (**b**), and the DEET-bound structure (**c**). Details can be found in Extended Data Table 1.

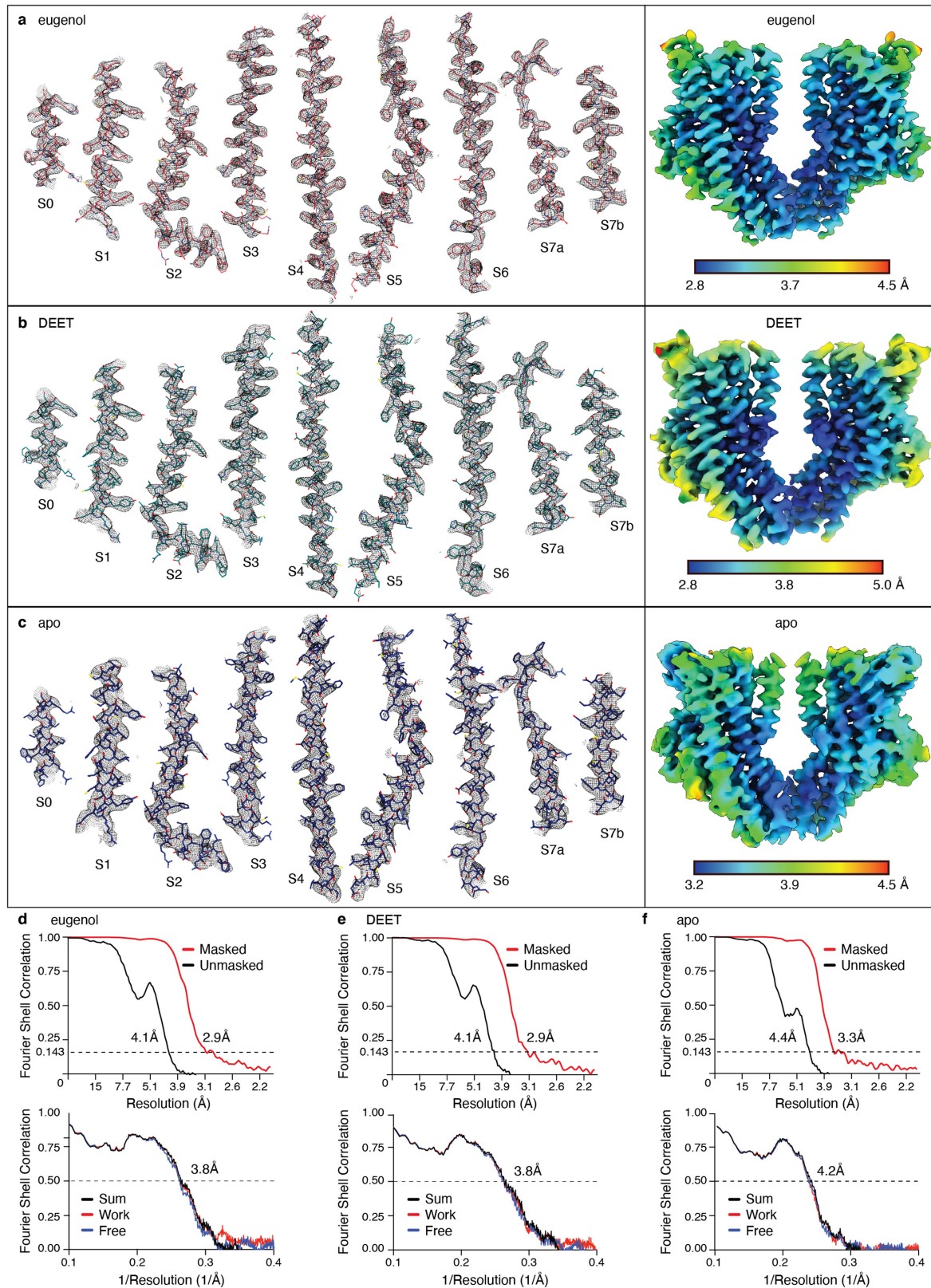

**Extended Data Fig. 4 | Cryo-EM density with map and model analysis and validation. a–c**, Left, cryo-EM densities for the modelled regions of the eugenol-bound structure (**a**), the DEET-bound structure (**b**), and the apo structure (**c**). Models are shown in stick representation within the density, with the helices denoted underneath from the N terminus (S0) to the C terminus (S7b). Right, local resolution estimation for each final map, calculated in cryoSPARC v2. Side views shown with front and back subunits removed for visualization. **d–f**, Top, FSC curves for the final cryo-EM density maps, obtained with cryoSPARC v2. The horizontal dashed line intersects at 0.143, the cutoff value. Bottom, FSC relationships between final map and model (black, sum), half-map 1 and model (red, work), and half-map 2 and model (blue, free), calculated in phenix.mtriage.

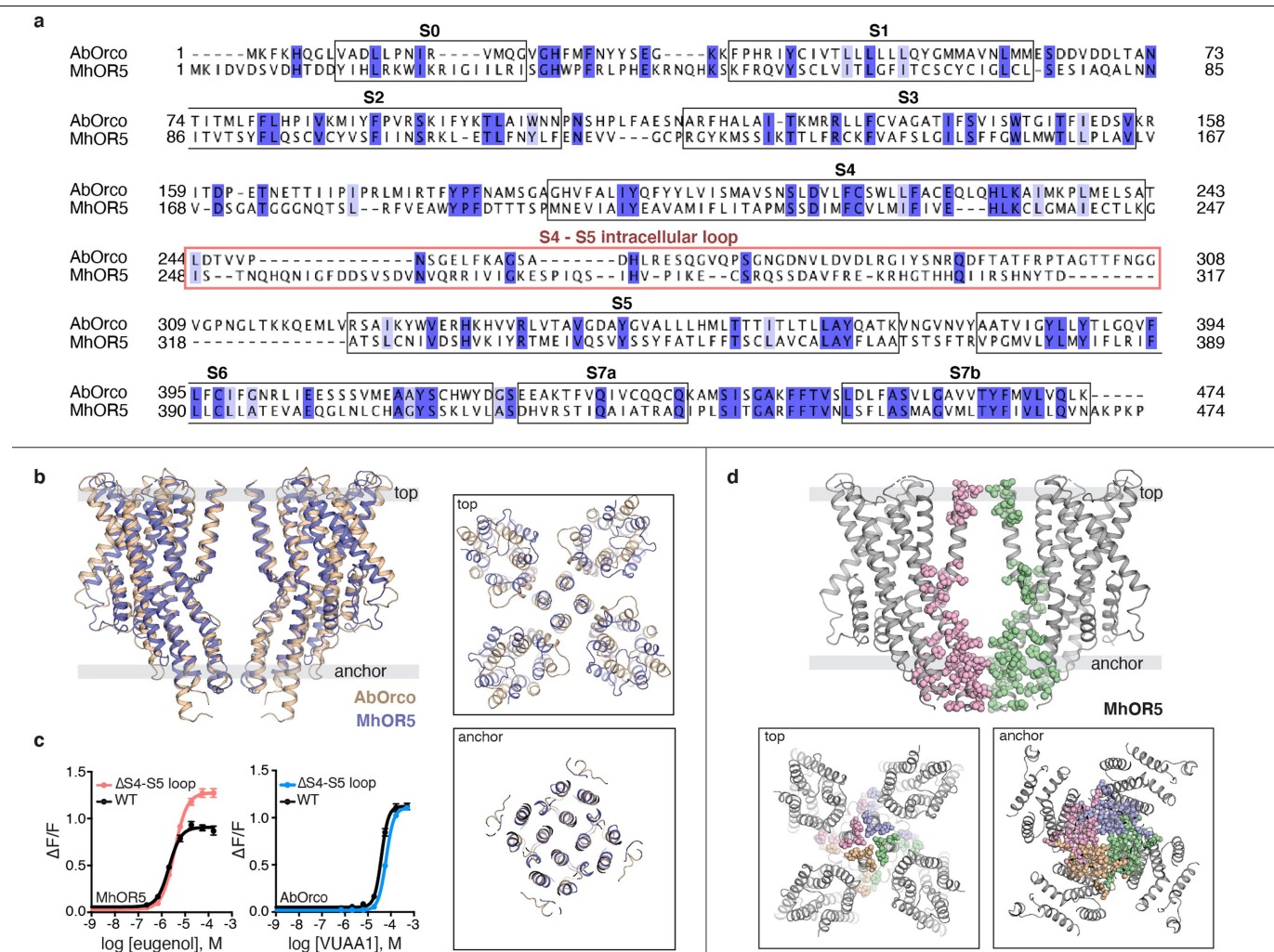

**a**, Sequence alignment showing positions S0–S7b with AbOrco and MhOR5 sequences, including the S4–S5 intracellular loop region (highlighted in red).

**Extended Data Fig. 5 | Conserved architecture of insect olfactory receptors. a**, Sequence alignment of *Mh*OR5 and *A. bakeri* Orco. Sequence identity (dark purple) and similarity (light purple) are highlighted. The positions of the helices in *Mh*OR5 are marked, as well as the S4–S5 intracellular loop. **b**, Structural overlay of *Ab*Orco (gold) and the apo state of *Mh*OR5 (blue) from the side view (left) with grey bars indicating the positions at which cross-sections are taken for insets shown from top views (top right) and anchor views (bottom right). The r.m.s.d. between individual subunits is 3.00 Å, and superposition of the whole tetramers yields an r.m.s.d. of 3.18 Å (calculated using PDBeFold server). **c**, Dose–response curves of wild-type and mutant *Mh*OR5 (left) and *Ab*Orco (right), in which the S4–S5 intracellular loop (unmodelled in both *Mh*OR5 and *Ab*Orco structures) has been replaced by a short linker (ΔS4–S5 loop), in response to eugenol (*n* = 6) and VUAA1 (*n* = 4), respectively. Error bars denote s.e.m. **d**, Inter-subunit interactions are concentrated in the anchor region. Top, side view of *Mh*OR5 with front and back subunits removed for visualization. Residues within 5 Å of residues in neighbouring subunits are shows as spheres, coloured by subunit. Insets (below) show extracellular views of cross-sections taken at the top and anchor positions as indicated by grey bars in the side view.

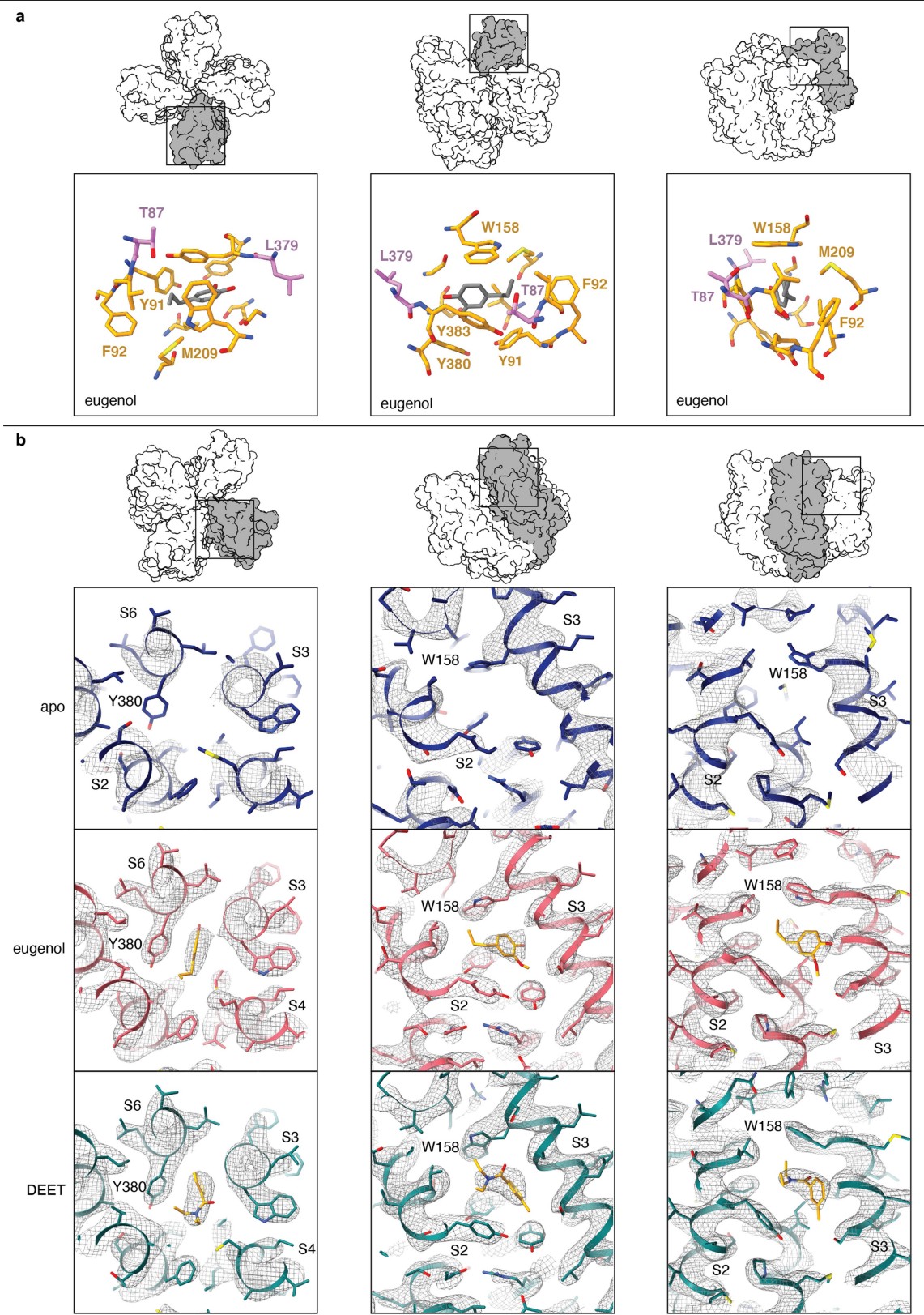

**Extended Data Fig. 6 | Detail of the odorant-binding region in *Mh*OR5.**
**a**, The binding pocket and ligand are shown in three orientations. The position of the pocket in each panel view is shown embedded within the tetrameric receptor to aid with orientation. Residues are colour-coded as in the bar graph of Fig. 3c, displaying the position of the 'control' residues L379 and T87 in purple that project away from the binding site and the binding site residues in yellow. The eugenol molecule is shown in grey. **b**, Schematic of the position of the three different views of the binding pocket. Model shown as ribbon and density shown as black mesh of the odorant-binding region of the apo (top), eugenol-bound (middle) and DEET-bound (bottom) structures. Cryo-EM density contoured at same level in all panels.

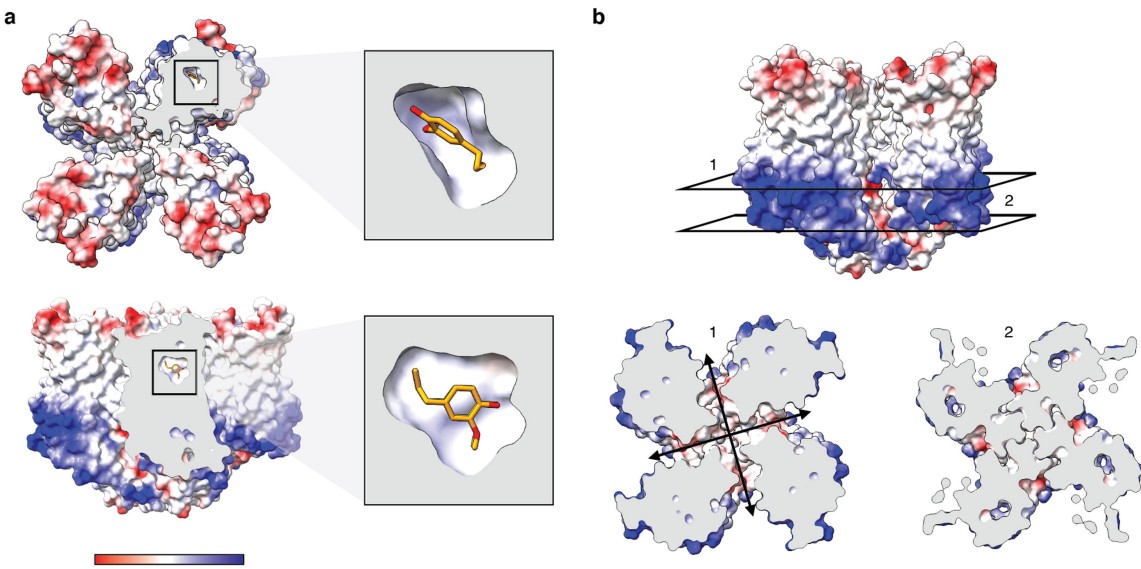

-10 kcal/mol*e 10

**Extended Data Fig. 7 | Electrostatic surface representation of the eugenol-bound model. a**, Top and side views of *Mh*OR5 bound to eugenol, with detailed views of the ligand binding cavity. **b**, Cross-sections of eugenol-bound *Mh*OR5 at the positions indicated with planes, highlighting the electrostatic environment of the side exits and the occlusion at the level of the anchor domain. Potentials estimated in ChimeraX v1.1 using coulombic calculation.

**Extended Data Fig. 8 | Docking of *Mh*OR5 agonists in the binding pocket.**
**a**, View of the binding pocket in the eugenol-bound (top, pink) and DEET-bound (middle, teal) *Mh*OR5 structures with the representative top poses for eugenol and DEET. Cryo-EM densities for eugenol and DEET shown as black mesh. The docking scores (calculated using Glide) of the eugenol poses (top) are, from left to right, −6,59, −6.71 and −6,81; and of the DEET poses (middle) are −7.33 (left) and −7.37 (right). Bottom, the top poses for each of the agonists of *Mh*OR5 from Glide are presented overlaid according to chemical class. **b**, Chemical structures of all activators in **a**.

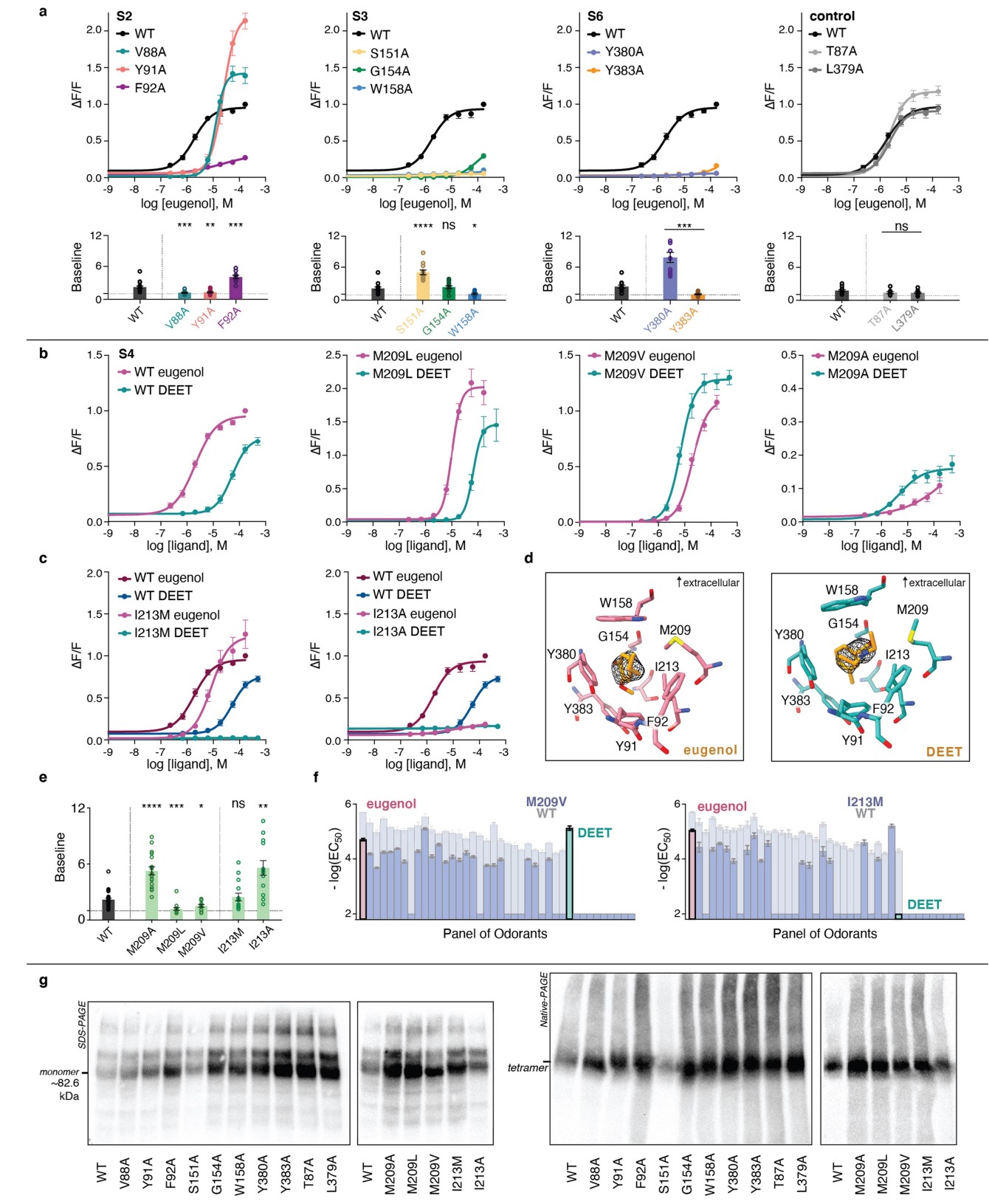

**Extended Data Fig. 9** | See next page for caption.

**Extended Data Fig. 9 | Effects of binding pocket mutations on *Mh*OR5 function. a**, Top, dose–response curves of *Mh*OR5 mutants in S2, S3 and S6 helices and two residues adjacent to the binding pocket but projecting away from it (denoted as control). Each WT curve represents the corresponding controls from the same experiments. All dose–response curves represent the average across experiments, error bars denote s.e.m. Bottom, baseline fluorescence of each mutant normalized to GCaMP-only control on the same plate. For all baseline data, columns are presented as mean values ± s.e.m. and statistical significance was determined using one-way ANOVA followed by Dunnett's multiple comparison tests comparing mutants to the respective wild-type *Mh*OR5 control experiments. For mutants where the $EC_{50}$ was incalculably high and Bartlett's test showed non-homogenous variance, statistical significance was determined with a Brown–Forsythe test. **b**, **c**, Dose–response

curves for WT, M209 mutant series and I213 mutant series of *Mh*OR5 in response to eugenol (pink) and DEET (teal). **d**, Detailed views of the binding pocket in the eugenol-bound (left, pink) and DEET-bound (right, teal) *Mh*OR5 structures. **e**, For each mutant shown in **b**, **c**, baseline fluorescence of each mutant normalized to GCaMP-only control on the same plate. **f**, Tuning curves of M209V, I213M and WT *Mh*OR5 showing −log($EC_{50}$), ordered as in Fig. 4e. **g**, Western blots of denaturing and native gels showing protein expression and multimeric assembly for each mutant, stained with anti-GFP. Full lanes shown and all gels run in parallel from samples in the same experiment. For all baseline measurements, baseline represents the mean of the first 30 s of fluorescence before the addition of any ligand. Relevant summary tables, including *n* values for biological replicates, are in Supplementary Tables 2, 4. ****$P < 0.0001$; ***$P < 0.0005$; **$P < 0.005$, *$P < 0.05$, NS, not significant.

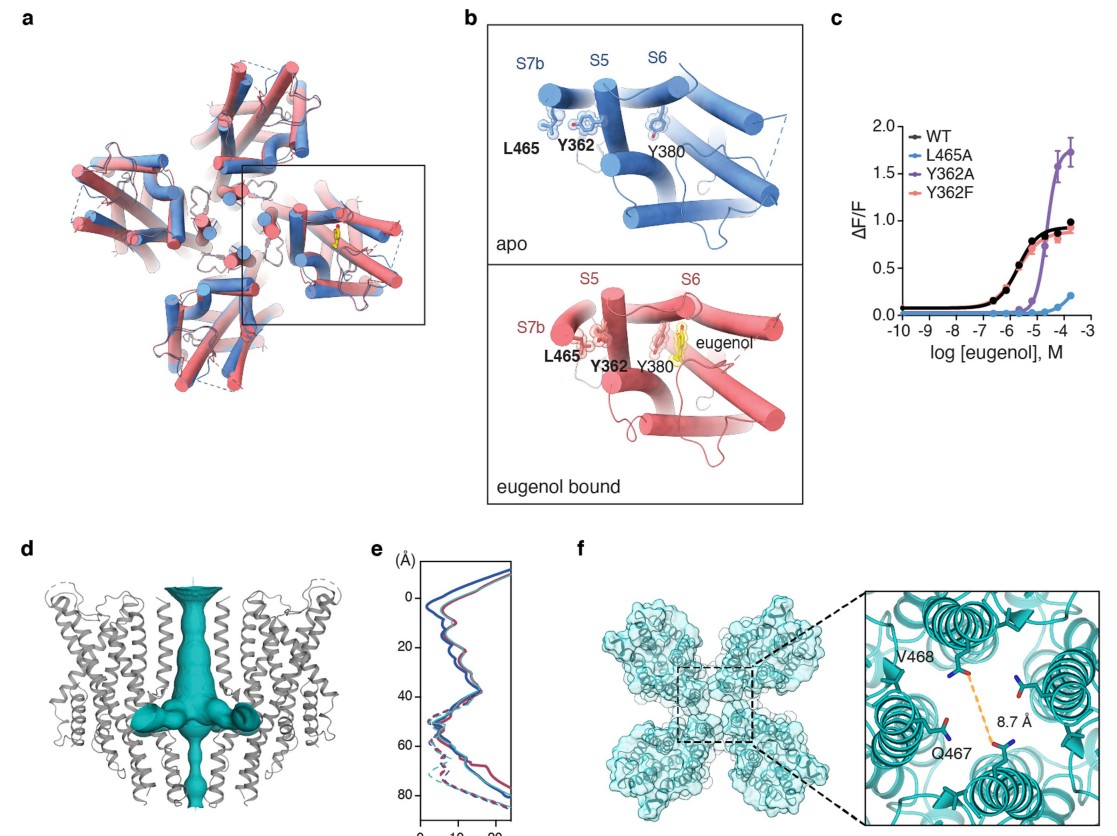

**Extended Data Fig. 10 | A potential route for coupling of odorant binding to pore opening and details of pore opening in the DEET-bound *Mh*OR5 structure. a**, Top view of *Mh*OR5 with helices represented as tubes in apo structure (blue) and eugenol-bound structure (pink). **b**, Close-up view of one subunit of the apo (blue, top) and eugenol-bound (pink, bottom) structures of *Mh*OR5. Residues Leu465 (in S7b), Tyr362 (in S5) and Tyr380 (in S6, lining the binding pocket) and eugenol are shown as sticks with a translucent outline of the sphere representation. **c**, Mutation of Leu465 in S7 and Tyr362 in S5 into alanine impairs receptor function. A conservative substitution of Tyr362 to phenylalanine restores wild-type activity, highlighting the potential role of hydrophobic

packing in connecting odorant binding with pore opening (*n* = 6). Dose–response curves represent the average across experiments, error bars denote s.e.m. Summary table of receptor data is in Supplementary Table 2. **d**, The lumen of the central pathway and side exits of the DEET-bound structure of *Mh*OR5. **e**, Diameter of the ion pathway beginning at the extracellular membrane (position 0) and following the ion conduction pathway (solid line) and the central four-fold axis through the anchor domain (dashed line), calculated with HOLE. Blue line represents apo structure, pink line represents eugenol-bound structure and cyan line represents DEET-bound structure. **f**, Top view of the DEET-bound structure with inset (right) highlighting the positions of residues Gln467 and Val468.

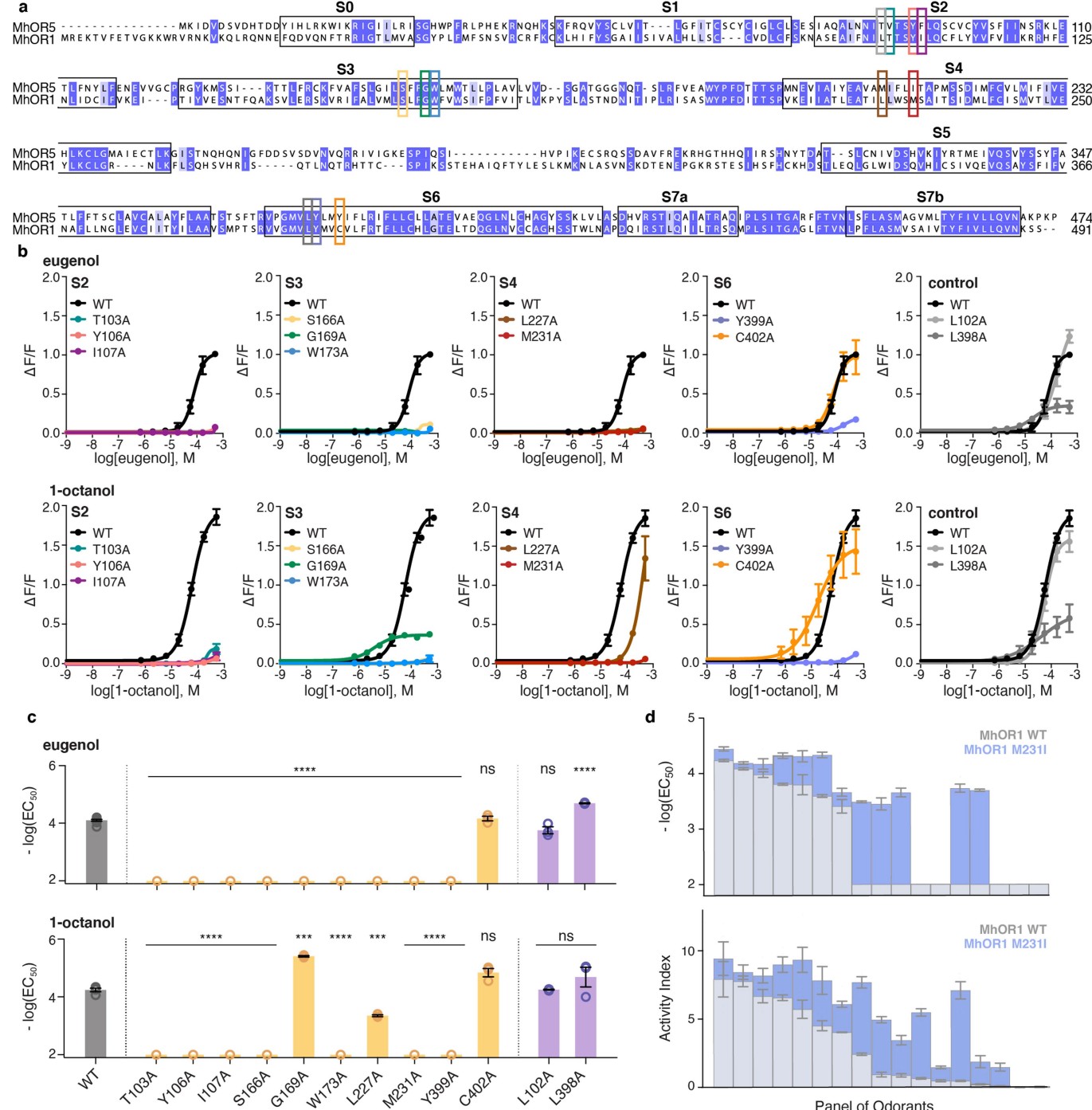

**Extended Data Fig. 11 | Putative binding pocket in *Mh*OR1. a**, Sequence alignment of *Mh*OR5 and *Mh*OR1, highlighting the position of the S0–S7 helices in *Mh*OR5 and the location of the binding pocket residues in *Mh*OR5 and homologous residues in *Mh*OR1, colour-coded as in **b**. **b**, Dose–response curves of *Mh*OR1 mutants (*n* = 3). Curves for mutants T103A, Y106A and I107A with eugenol, G169A and W173A with eugenol, and L227A and M231A with eugenol overlap almost completely and therefore appear overlaid. Dose–response curves represent the average across experiments, error bars denote s.e.m. **c**, Mutations of putative binding pocket residues in *Mh*OR1 significantly and differentially alter ligand binding for the two best *Mh*OR1 ligands, 1-octanol and eugenol. Mutation of the putative 'control' residues in *Mh*OR1 (homologous to those that in *Mh*OR5 are adjacent to the pocket but projecting

away from the ligand) are also not directly involved in ligand binding in *Mh*OR1 (*n* = 3). Statistical significance determined using one-way ANOVAs followed by Dunnett's multiple comparison tests, comparing mutants to their respective wild-type controls for each ligand. For mutants where the $EC_{50}$ was incalculably high and Bartlett's test showed non-homogenous variance, statistical significance was determined with a Brown–Forsythe test. ****$P < 0.0001$; ***$P < 0.001$; NS, not significant. Data shown as mean ± s.e.m. **d**, $EC_{50}$ and activity index tuning curves for wild-type *Mh*OR1 and a conservative binding pocket mutant, M231I, against 17 diverse ligands. Ligands are ordered high-to-low by $EC_{50}$ and activity index scores for the wild-type *Mh*OR1. Data shown as mean ± s.e.m. Relevant summary tables, including *n* values for the biological replicates, can be found in Supplementary Tables 10, 11.

**Extended Data Table 1 | Cryo-EM data collection, refinement and model statistics**

| | apo<br>EMDB-22372<br>PDB 7LIC | eugenol<br>EMDB-22374<br>PDB 7LID | DEET<br>EMDB-22375<br>PDB 7LIG |
|---|---|---|---|
| **Data colelction and processing** | | | |
| Magnification | 29000x | 29000x | 29000x |
| Voltage (kV) | 300 | 300 | 300 |
| Electron exposure (e-/^2/frame) | 1.5 | 1.22 | 1.3 |
| Defocus range (μm) | -1 to -3 | -1 to -3 | -1 to -3 |
| Pixel Size (Å) | 1.03 | 1.03 | 1.03 |
| Symmetry imposed | C4 | C4 | C4 |
| Initial particle images (no.) | 1,098,939 | 461.254 | 787,448 |
| Final particle images (no.) | 49,832 | 53,900 | 56,190 |
| Map resolution (Å) | 3.3 | 2.9 | 2.9 |
| FSC threshold | 0.143 | 0.143 | 0.143 |
| Map resolution range (Å) | 3.3 - 4.5 | 2.8 - 4.5 | 2.9 - 5.0 |
| | | | |
| **Refinement** | | | |
| Initial model used (PDB code) | 6C70 | 6C70 | 6C70 |
| Model resolution (Å) | 4.0 | 4.0 | 4.0 |
| FSC threshold | 0.5 | 0.5 | 0.5 |
| Map sharpening $B$ factor (Å^2) | -121.8 | -99 | -99.9 |
| Model composition | | | |
| Non-hydrogen atoms | 11,980 | 12,092 | 12,092 |
| Protein residues | 377 | 381 | 381 |
| Ligands | 0 | 1 | 1 |
| $B$ factors (Å^2) | | | |
| Protein | 121.19 | 39.04 | 59.08 |
| Ligand | N/A | 34.11 | 55.07 |
| R.m.s. deviations | | | |
| Bond lengths (Å) | 0.004 | 0.005 | 0.005 |
| Bond angles (°) | 0.8 | 0.833 | 0.785 |
| Validation | | | |
| MolProbity score | 1.28 | 1.5 | 1.5 |
| Clashscore | 5.27 | 3.54 | 4.28 |
| Poor rotamers (%) | 0.9 | 1.47 | 1.47 |
| Ramachandran plot | | | |
| Favored (%) | 98.64 | 96.53 | 97.07 |
| Allowed (%) | 1.36 | 3.47 | 2.93 |
| Disallowed (%) | 0 | 0 | 0 |

**Extended Data Table 2 | The correlation between chemical metrics and receptor activity for *Mh*OR5 and *Mh*OR1**

| MhOR5 | Polar Surface Area (A^2) | H-bond total | H-bond acceptor | Water Solubility (g/L) | H-bond donor | XlogP3 | Polariz ability (A^3) | Rotatable bond count | Vapor Pressure (mm Hg) | Molecular Weight (g/mol) | Area (A^2) |
|---|---|---|---|---|---|---|---|---|---|---|---|
| PSA (A^2) | 0.28 | | | | | | | | | | |
| H-bond total | 0.28 | 0.24 | | | | | | | | | |
| H-bond acceptor | 0.28 | 0.25 | 0.23 | | | | | | | | |
| Water Solubility (g/L) | 0.30 | 0.27 | 0.28 | 0.22 | | | | | | | |
| H-bond donor | 0.28 | 0.25 | 0.25 | 0.23 | 0.16 | | | | | | |
| XlogP3 | 0.29 | 0.24 | 0.23 | 0.22 | 0.18 | 0.12 | | | | | |
| Polarizability (A^3) | 0.28 | 0.24 | 0.23 | 0.22 | 0.16 | 0.13 | 0.02 | | | | |
| Rotatable bond count | 0.35 | 0.29 | 0.27 | 0.30 | 0.21 | 0.29 | 0.06 | 0.02 | | | |
| Vapor Press. (mm Hg) | 0.31 | 0.27 | 0.24 | 0.25 | 0.21 | 0.12 | 0.03 | 0.04 | 0.02 | | |
| MW (g/mol) | 0.28 | 0.24 | 0.23 | 0.24 | 0.16 | 0.15 | 0.26 | 0.02 | 0.03 | 0.01 | |
| Area (A^2) | 0.28 | 0.24 | 0.23 | 0.24 | 0.16 | 0.16 | 0.06 | 0.04 | 0.02 | 0.03 | 0.00 |

| MhOR1 | Polar Surface Area (A^2) | H-bond total | H-bond acceptor | Water Solubility (g/L) | H-bond donor | XlogP3 | Polariz ability (A^3) | Rotatable bond count | Vapor Pressure (mm Hg) | Molecular Weight (g/mol) | Area (A^2) |
|---|---|---|---|---|---|---|---|---|---|---|---|
| PSA (A^2) | 0.05 | | | | | | | | | | |
| H-bond total | 0.06 | 0.06 | | | | | | | | | |
| H-bond acceptor | 0.14 | 0.10 | 0.10 | | | | | | | | |
| Water Solubility (g/L) | 0.05 | 0.06 | 0.10 | 0.03 | | | | | | | |
| H-bond donor | 0.05 | 0.10 | 0.10 | 0.03 | 0.02 | | | | | | |
| XlogP3 | 0.05 | 0.06 | 0.10 | 0.04 | 0.03 | 0.03 | | | | | |
| Polarizability (A^3) | 0.05 | 0.06 | 0.10 | 0.03 | 0.02 | 0.03 | 0.00 | | | | |
| Rotatable bond count | 0.10 | 0.11 | 0.14 | 0.08 | 0.06 | 0.13 | 0.05 | 0.03 | | | |
| Vapor Press. (mm Hg) | 0.07 | 0.09 | 0.11 | 0.06 | 0.05 | 0.04 | 0.02 | 0.05 | 0.02 | | |
| MW (g/mol) | 0.05 | 0.06 | 0.10 | 0.04 | 0.02 | 0.04 | 0.11 | 0.03 | 0.03 | 0.00 | |
| Area (A^2) | 0.06 | 0.07 | 0.10 | 0.04 | 0.02 | 0.05 | 0.05 | 0.03 | 0.03 | 0.01 | 0.00 |

Multiple regression analysis between pairs of chemical descriptors and receptor activity for *Mh*OR5 (top) and *Mh*OR1 (bottom). Values in each cell are the $R^2$ values of the regression analysis between the descriptors in the respective column and row, combined, and the receptor's activity index in response to a panel of 54 ligands. Diagonal cells reflect the $R^2$ values for a simple linear regression between the corresponding descriptor and activity index of that receptor. For both receptors, polar surface area, hydrogen bond count, water solubility, vapour pressure, rotatable bond count and molecular weight, individually, correlate inversely with receptor activity, and XlogP3 and polarizability correlate positively. Note correlations are higher for *Mh*OR5 than for *Mh*OR1. Molecular descriptors used are compiled in Supplementary Table 9.

# nature research

# Reporting Summary

Nature Research wishes to improve the reproducibility of the work that we publish. This form provides structure for consistency and transparency in reporting. For further information on Nature Research policies, see our Editorial Policies and the Editorial Policy Checklist.

## Statistics

For all statistical analyses, confirm that the following items are present in the figure legend, table legend, main text, or Methods section.

| n/a | Confirmed | |
|---|---|---|
| ☐ | ☒ | The exact sample size (*n*) for each experimental group/condition, given as a discrete number and unit of measurement |
| ☐ | ☒ | A statement on whether measurements were taken from distinct samples or whether the same sample was measured repeatedly |
| ☐ | ☒ | The statistical test(s) used AND whether they are one- or two-sided<br>*Only common tests should be described solely by name; describe more complex techniques in the Methods section.* |
| ☒ | ☐ | A description of all covariates tested |
| ☐ | ☒ | A description of any assumptions or corrections, such as tests of normality and adjustment for multiple comparisons |
| ☐ | ☒ | A full description of the statistical parameters including central tendency (e.g. means) or other basic estimates (e.g. regression coefficient) AND variation (e.g. standard deviation) or associated estimates of uncertainty (e.g. confidence intervals) |
| ☐ | ☒ | For null hypothesis testing, the test statistic (e.g. *F*, *t*, *r*) with confidence intervals, effect sizes, degrees of freedom and *P* value noted<br>*Give P values as exact values whenever suitable.* |
| ☒ | ☐ | For Bayesian analysis, information on the choice of priors and Markov chain Monte Carlo settings |
| ☒ | ☐ | For hierarchical and complex designs, identification of the appropriate level for tests and full reporting of outcomes |
| ☐ | ☒ | Estimates of effect sizes (e.g. Cohen's *d*, Pearson's *r*), indicating how they were calculated |

*Our web collection on statistics for biologists contains articles on many of the points above.*

## Software and code

Policy information about availability of computer code

| | |
|---|---|
| Data collection | For cryo-EM collection: SerialEM; for electrophysiology data: Clampex 10.6. |
| Data analysis | Relion-3.0, cryoSPARCv2, PyMOL, Chimera, ChimeraX, Schrodinger Maestro, GraphPad Prism, Coot, MotionCor2, CTFFIND4, PHENIX, HOLE, JalView |

For manuscripts utilizing custom algorithms or software that are central to the research but not yet described in published literature, software must be made available to editors and reviewers. We strongly encourage code deposition in a community repository (e.g. GitHub). See the Nature Research guidelines for submitting code & software for further information.

## Data

Policy information about availability of data

All manuscripts must include a data availability statement. This statement should provide the following information, where applicable:

- Accession codes, unique identifiers, or web links for publicly available datasets
- A list of figures that have associated raw data
- A description of any restrictions on data availability

The 3D cryo-EM density map of unbound MhOR5, eugenol-bound MhOR5, and DEET-bound MhOR5 have been deposited in the Electron Microscopy Data Bank under accession numbers EMD-23372, EMD-23374, and EMD-23375, respectively. The coordinates of the atomic models of unbound MhOR5, eugenol-bound MhOR5, and DEET-bound MhOR5 have been deposited in the Protein Data Bank under accession numbers 7LIC, 7LID and 7LIG, respectively. Raw data associated with functional analyses in Fig. 1c,2e,2f,3d,4d,4e and Extended Data Fig. 1d,1f,2,10a-c,11c is available upon request and summarized in Tables 2,4,6-9.

April 2020

# Field-specific reporting

Please select the one below that is the best fit for your research. If you are not sure, read the appropriate sections before making your selection.

☒ Life sciences ☐ Behavioural & social sciences ☐ Ecological, evolutionary & environmental sciences

For a reference copy of the document with all sections, see nature.com/documents/nr-reporting-summary-flat.pdf

# Life sciences study design

All studies must disclose on these points even when the disclosure is negative.

| | |
|---|---|
| Sample size | No calculations were performed to determine sample sizes; however, the addition of more data did not alter conclusions from this study. |
| Data exclusions | For processing cryo-EM data, some particles were excluded following standard procedures and described extensively in the methods section. |
| Replication | Functional experiments were repeated on different days, using independently transfected cells and independently prepared odorant solutions. Cryo-EM datasets were collected on 4 different days: the apo structure comes from 2 datasets collected on independent days, and the eugenol and DEET structures come from one single dataset each. |
| Randomization | This study did not allocate experimental groups; thus, no randomization was necessary. |
| Blinding | No blinding was used; all functional data were analyzed using the same methods and all data were included in the results. |

# Reporting for specific materials, systems and methods

We require information from authors about some types of materials, experimental systems and methods used in many studies. Here, indicate whether each material, system or method listed is relevant to your study. If you are not sure if a list item applies to your research, read the appropriate section before selecting a response.

### Materials & experimental systems

| n/a | Involved in the study |
|---|---|
| ☒ ☐ | Antibodies |
| ☐ ☒ | Eukaryotic cell lines |
| ☒ ☐ | Palaeontology and archaeology |
| ☒ ☐ | Animals and other organisms |
| ☒ ☐ | Human research participants |
| ☒ ☐ | Clinical data |
| ☒ ☐ | Dual use research of concern |

### Methods

| n/a | Involved in the study |
|---|---|
| ☒ ☐ | ChIP-seq |
| ☒ ☐ | Flow cytometry |
| ☒ ☐ | MRI-based neuroimaging |

# Eukaryotic cell lines

Policy information about cell lines

| | |
|---|---|
| Cell line source(s) | Sf9 (ATCC CRL-1711). HEK293S GnTi- (ATCC CRL-3022). |
| Authentication | None were authenticated. |
| Mycoplasma contamination | Cells were tested for mycoplasma and no contamination was detected. |
| Commonly misidentified lines (See ICLAC register) | N/A. |

