## [Peer Review File · Nature]

Manuscript Title: The structural basis of odorant recognition in insect olfactory receptors

Reviewer Comments & Author Rebuttals

Reviewer Reports on the Initial Version:

Referee expertise:

Referee #1: sensory receptors

Referee #2: cryo-EM, channels

Referee #3: ORs

Referees' comments:

Referee #1 (Remarks to the Author):

This manuscript by del Marmol et al. reports the structure of an insect odorant receptor in both closed (apo) and ligand-bound/open conformations, allowing the authors to make important and novel conclusions regarding the mechanisms of ligand binding and ion permeation. Insect odorant receptors comprise possibly the largest family of ion channels, yet a basic understanding of their functions and structures has lagged behind that of most other ion channel families. In fact, until the Ruta lab published the structure of the Orco subunit of the odorant receptors, there was still some doubt that they formed bonafide ion channels. This work did not, however, resolve the open state of a ligand-bound channel, as the Orco subunit does not bind ligand found naturally (the channels typically assemble as heteromultimers). Now the Ruta lab has overcome this hurdle through a heroic effort to identify a receptor that functions as a homomultimer of subunits that open in response to ligand binding (MhOR5), and then solve its structure in apo and ligand-bound state. This allows them to resolve a clear transition in opening of the MhOR5 receptor in which the S7b helices rotate so that a hydrophobic residue (Val468) moves out of the pore lumen and it is replaced by a polar residue (Gln467). This movement also widens the pore, to a size (9.2 Å) that can accommodate a hydrated cation. Interestingly, Gln467 is highly conserved across all insect ORs, and mutation of this residue (to alanine or arginine, but not asparagine) severely impairs function. They also identified a common binding site for two activators - eugenol and DEET - and used site-directed mutagenesis to corroborate their findings and gain further insight into ligand binding and selectivity. Their results show that the remarkably broad tuning of the receptor can be attributed to the nature of a binding site which can accommodate ligands with very different structures. Overall this manuscript describes a significant body of work that greatly advances the field and will be of wide interest to scientists ranging from structural biologists, neuroscientists, and even ecologists.

1. I wonder if the authors tried mutating Val468 to glutamine (they report that mutating Val468 to asparagine is without affect). If this channel is functional, it would show that it is not just the presence of the glutamine within the pore that is important to promote ion conduction, but that the protein helix needs to rotate. (this could be concluded from the mutation to asparagine, but it is a little less direct).
2. Patch-clamp data shown in Figure 1D appears to represent a single recording from either whole-cell or single-channel recording mode. It would be good to have average data (including untransfected controls).
3. The use of one aggregate score, the "Activity index" to describe the potency of the ligands for this receptor (e.g. Fig 1c) loses essential information regarding the EC50 and the maximal efficacy for the ligands. This information is plotted in the supplementary figures, so the authors might simply consider moving it to the main figures. The same is true of Fig 4 (EC50 could be plotted as well as activity index).
4. It would be helpful if the authors would explain the supplementary data how MhOR5 was selected for these studies. In particular, which other receptors/species, if any, were tested.

Referee #2 (Remarks to the Author):

In the manuscript "The structural basis of odorant recognition in insect olfactory receptors", del Marmol et al. present a comprehensive structural and functional study of how insect olfactory receptors recognize odorants and relay this signal to the cell. Whereas vertebrates rely on GPCR-based signalling cascades, insects have evolved odorant-gated ion channels. The main question the study wants to answer is how a rather small number of odorant receptors can recognize a very large number of different odorants by combinatorial, promiscuous binding.

The authors decided to study odorant receptors of the jumping bristletail as this organism possesses a simplified system of five receptors which combine odorant recognition and ion channeling in the same polypeptide chain. Initially, they established odorant recognition profiles of two of those five, and then continued to study the more promiscuous one, MhOR5, in detail. They obtained and compared Cryo-EM structures of MhOR5 alone and in complex with the most potent activator eugenol as well as the insect repellent DEET. Strikingly, they found that the four S7b helices of the MhOR5 tetramer which compose the central ion pore turn upon ligand binding, thereby widening the pore and exposing more hydrophilic residues which are key to ion translocation. Odorant binding appears to take place in a single hydrophobic pocket in each of the four subunits which is rather far away from the pore. By analysis of the structures, mutational studies and *in silico* docking experiments with a larger number of odorants, the authors establish that ligand binding does not rely on single specific interactions with certain features/chemical groups of the ligands, but rather recognizes the overall shape of the compounds by flexible hydrophobic interactions, thereby achieving the observed promiscuity.

I believe that the results presented in this manuscript will be of great interest for the broad readership of Nature as they present a big advance towards understanding the molecular basis of odorant recognition in insects. The study is well carried out, data presented in a clear and logical way, the manuscript written such that also non-experts from outside the field can immediately follow the argumentation. Hence, I would recommend to accept this manuscript for publication once the following points have been addressed by the authors.

Major points

1. While the conformational changes of MhOR5 upon eugenol binding are already visible in the present figures, this would be even much more obvious in one or several morph videos between the two states. Here, especially the movement of helix S7b could be shown very clearly, and it would also become even more apparent that the conformational changes close to the ligand-binding pocket are rather small.
2. I do not share the authors' absolute confidence in interpreting the observed lower resolution of S7b in the apo as compared to the eugenol-bound structure. First, judging only from Extended data Fig. 4, it is not completely obvious to me that the difference in local resolution of this particular helix is significantly higher than the difference in resolution over the whole molecule. Second, while I agree with the authors that it is very tempting to speculate that a lower resolution (equalling higher mobility) of the pore helix might be due to spontaneous openings of the channel, the data don't allow to draw this conclusion with certainty, and thus the authors might want to tone this part down a bit.
3. In the experiment shown in Fig. 2f, the authors compare the activation efficiency of VUAA1 on Orco with the efficiency of acetophenone on Orco+OR28. However, in order to draw firm conclusions about the effect of the Q742A mutant, the authors would have needed to use the same agonist for both complexes. Furthermore, the authors claim that the partial rescue of the effect of the Orco Q472A mutant upon co-expression with OR28 is due to a compensation of the mutant in the heteromeric assembly. However, based on the presented data and the cited references, the authors cannot be absolutely sure that both proteins form such a heteromeric complex. Even if they assemble in heteromeric complexes (and chances are high that they do), it is very likely that not a single heterotetrameric species, but rather several distinct species (4x OR, 3x OR + 1x Orco, etc) co-exist in the cells. In this situation, with the presented data the authors cannot rule out that a species consisting of 4x OR could be responsible for the observed activity and that, due to potential differences of expression levels between wt and mutant Orco, also the levels of insect 4x OR would differ, the conclusion of this experiment is overstated. Since I believe that appropriate control experiments can be done with reasonable effort, I would highly

recommend that the authors perform them and present the results in a revised manuscript.

4. The authors present a conclusive analysis of MhOR5 and its binding specificity, but they do not provide a detailed comparison with the other MhORs or neopteran Orco or selected ORs from other species, even though they could probably at least deduce some insight into these proteins as well. Especially, it would be very interesting to know whether it is possible or even likely that these proteins harbor odorant pockets at the same position and how the volume and shape of these pockets differs from MhOR5. Even though the sequence similarity to these proteins is limited, the structures can be expected to be highly similar, so sequence alignments or homology models might give such initial insight. In the optimal case, such comparison might explain the higher selectivity of MhOR1, and even allow some predictions about the selectivity of the other three MhORs (however, this might be beyond the scope of this study).

5. Based on the structure and alignment data, it should be possible to predict whether the formation of stable heteromeric complexes between the MhORs is likely, and it would be interesting to hear the authors' opinion on whether such complexes would be biochemically stable and physiologically relevant.

Minor points:

1. The definition of the abbreviation 'ORs' is inconsistent. Once it is described as olfactory receptors (p.1 l.3) and another time as odorant receptor (p.2 l.31).
2. The introduction of pore architecture (p.5 l.145-148) would benefit from indicating the membrane in the corresponding Fig. 2a.
3. The authors should be more careful in their wording regarding values reported for distance of atoms (or rather atom centers) as for example on p. 5, l. 152, and the pore diameter, calculated by HOLE and displayed in Fig. 2b, that takes into account the van-der-Waals radii of the atoms.
4. P. 8, l. 242: Topologically, S5 runs parallel to S7b, but anti-parallel to S6.
5. On p.15, l. 520, a link is broken
6. P. 15, l. 543: 'After 30 s of a baseline recording' should be 'After 30 s of baseline recording'.

Referee #3 (Remarks to the Author):

This manuscript by del Marmol, Yedlin and Ruta describes structural and functional analyses of an odorant receptor from the jumping bristletail, MhOR5. They find that it is a homotetrameric ion channel with high structural similarity to the Orco structure recently published by the same lab. Their three structures provide insight into the gating-related conformational changes, with a closed-pore apo structure and two open-pore liganded structures. The two ligands they use, eugenol and DEET, are quite different in chemical properties, although similar in size, and they bind to the same site, deep in the middle of each subunit. They also provide data showing that the ligand selectivity of MhOR5 is quite broad, and contrast it to that of MhOR1, which is much more selective. They also introduce structure-based mutations in MhOR5 and demonstrate that these mutations can readily tune its selectivity in a way that can be explained with the structural information.

Overall, this is a well-designed and important study, well written with a balanced discussion of the results and an effective presentation throughout. Its insights are far-reaching in terms of how we understand how Ors work, but also how odor perception works more generally. Furthermore, the insights are not limited to just the field of odor perception in insects and arthropods; many of the lessons likely extend, at least conceptually, to odor sensing by mammalian G-protein coupled receptors.

I only have one significant concern with the analysis and presentation of the ligand-binding site data, which I believe can be readily addressed without significantly impacting the conclusions the authors make. I also have a few additional suggestions detailed below.

Of the mutations to the ligand binding pocket residues, the authors conclude on p. 7 that "mutation of any of these residues to alanine strongly altered eugenol signaling". While that is technically the case, the data they show do not rule out issues with protein expression, folding or trafficking, rather than alteration of the function of the ligand-binding pocket. While they do show protein expression data in Ext. Fig. 10d, it is unclear whether these strips come from a single gel image. Also, is this a western blot? The legend is unclear. Even assuming that this is a single gel image, there are variations in protein expression, and folding or trafficking were not tested.

Furthermore, the methods suggest that these data are from a different protein construct (GFP-tagged) transfected using a different protocol than the one used for the functional assays. It would be much preferable to at least (1) show a single gel image with all mutants compared to WT, and (2) use the same transfection protocol as for the functional assays, even if the authors must use a different construct for western blot detection purposes.

The other related issue I have with the data analysis is the use of the Activity Index to compare WT and mutant constructs (Figs 2e-f, 3d, 4d-e). (To be clear, I think that the Activity Index is a useful metric to compare the activity of different ligands on the same protein.) Because the Activity Index incorporates the $\Delta F/F$ ratio, it depends strongly on the basal activity of the ion channels; mutated MhOR5 with high basal activities will have a reduced Activity Index, but that does not necessarily indicate a reduced sensitivity to the odorant. Similarly, mutated MgOR5s with lower expression levels (or trafficking to the plasma membrane) will have consequently reduced ΔF values. So, in comparing WT to mutants, the EC50 values are likely the better indicators of the sensitivity to specific ligands.

Related to the point above, while I appreciate that baseline subtractions in Ext. Fig. 10a-c help focus the attention on the EC50 changes, it would be useful to somehow indicate which mutations lead to elevated baselines (data in part (d)) in the legends to the dose-response curves.

Additional specific comments and questions:

On page 6, the authors state "with Trp158 forming the lid". This suggests that the ligand is buried and not solvent-accessible. Is that the case, and if so, is it occluded in the apo state as well? This would be worth noting in the main text. Also, in Figure 3c, is the general orientation such that the extracellular face is up? This would fit with my understanding of the "lid" description, and if that is the case, it would be worth noting in the legend.

Related to the comment above, it would be useful to show an electrostatics surface representation of the binding pocket, to provide another illustration of its chemical properties.

MhOR5 seems to consistently show multiple bands on SDS-PAGE western blots (Ext. Fig 1b and Ext Fig 10d). Is it because it is glycosylated? If so, are any of the sugars visible in the maps?

Methods: In general, the methods are nicely detailed. The one exception I found is that the description of each construct is lacking. The plasmid constructs used for protein purification and functional assays should be detailed enough that a reader can reconstruct the exact protein sequence at a minimum (and the exact DNA sequence, ideally). As is, on p. 13, I do not know for sure the order of the Strep II tag, GFP and 3C protease sites in the construct, for example.

On p. 14 the authors mention that caffeine docks favorably but does not activate. Does it function as a competitive inhibitor of eugenol (or DEET)?

The validation report for the apo structure includes a resolution estimate of 3.64 Å based on the author-provided FSC curve (p. 21), rather than the 3.30 Å reported by the authors. This lower resolution seems more in line with the much lower map quality in comparison to the two liganded structures. Can the authors adjust their estimate, or justify why 3.30 Å is more appropriate? (This does not detract from the authors' conclusions, and they were very careful in their discussion of the map quality.)

In Figures 1c, Ext. Fig. 1, and Ext. Fig. 2 the authors use a centered tuning curve, whereas in Figure 4e, they use a left-to-right tuning curve. I find the left-to-right presentation easier to take in. And particularly in Ext. Fig. 2 the associated dose-response panels in (d) are then organized with similarly activating ligands far apart in the figure.

In Figure 1e-f, the disordered loops are indicated with dotted lines. But it wasn't until I looked at the validation report that I realized that one of these loops (I believe it is the S4-S5 loop?) is 70-residues long! Looking at the two sequence alignments included in the manuscript, this loop is also long in both MhOR1 and AbOrco, but poorly conserved. A note about this loop in the section describing the overall structure would be useful.

The description of the pores and their illustrations in Figure 2a-c and Ext. Fig. 12 leave me curious about the electrostatics properties of the pore surfaces, especially the side exits. Could the authors perhaps add some illustrations of the electrostatics in Ext. Fig. 12?

In Fig. 3, it would be helpful to color-code parts (c) and (d) similarly, with the milder phenotypes (V88A, Y91A) colored differently, and if possible, include the control residues in part (c) as well.

Ext. Fig. 1 (and Ext. Fig. 6): I would be useful to include a panel that illustrates the white-to-blue coloring of the sequence alignments on the MhOR5 structure.

Ext. Fig. 6: what is the RMSD for the tetramer-to-tetramer superposition? And for the subunit-to-subunit superposition? (In other words, are some of the large differences in helix position at the periphery due to a shift in the tetramer organization?)

Ext. Fig. 9: Can the authors include the chemical structures of the compounds in part (c)? (Structures of eugenol and DEET would also be useful for comparison.)

Minor notes:

On p. 15, the methods include "Error! Reference source not found."

In at least one place (legend to Ext. Fig. 12) the authors refer to the apo structure as the "unbound structure". It would be good to check the manuscript carefully.

Similarly, the methods refer to "Abak", which I gather is *A. bakeri*.

References 59 and 63 are duplicates.

Author Rebuttals to Initial Comments:

2021-01-1459A: The structural basis of odorant recognition in insect olfactory receptors

We thank the referees for their enthusiasm for our work and insightful suggestions, which have considerably improved the clarity, rigor, and scope of our work.

In the revised manuscript, we now demonstrate that the insights obtained from our studies of MhOR5 can shed light on the chemical tuning of other receptors by performing a functional analysis of the binding pocket of MhOR1, a more narrowly tuned receptor from the same species. We identified ten residues predicted to line the binding pocket of MhOR1 based on sequence homology with MhOR5, and validated these amino-acids play a critical role in ligand recognition using site-directed mutagenesis and functional calcium imaging. Moreover, we showed how a single conservative mutation in the binding pocket of MhOR1 could broadly reconfigure the receptor's chemical tuning to a panel of odorants, mirroring our observations from MhOR5. Together, these new results further support a model that the promiscuous tuning of ORs relies on shared structural determinants within a common binding site, even for receptors that display distinct chemical tuning. In addition, we expanded our electrophysiological analysis, to provide further quantification of the basic functional characteristics of MhOR5 and included appropriate controls for these experiments and for the expression of mutant channels. We characterized the electrostatic environment of the binding pocket and side exits of MhOR5, modified and reorganized many figures to improve clarity. A detailed point by point response to the referees' remarks is below.

Referee #1 (Remarks to the Author):

This manuscript by del Marmol et al. reports the structure of an insect odorant receptor in both closed (apo) and

ligand-bound/open conformations, allowing the authors to make important and novel conclusions regarding the mechanisms of ligand binding and ion permeation. Insect odorant receptors comprise possibly the largest family of ion channels, yet a basic understanding of their functions and structures has lagged behind that of most other ion channel families. In fact, until the Ruta lab published the structure of the Orco subunit of the odorant receptors, there was still some doubt that they formed bonafide ion channels. This work did not, however, resolve the open state of a ligand-bound channel, as the Orco subunit does not bind ligand naturally (the channels typically assemble as heteromultimers). Now the Ruta lab has overcome this hurdle through a heroic effort to identify a receptor that functions as a homomultimer of subunits that open in response to ligand binding (MhOR5), and then solve its structure in apo and ligand-bound state. This allows them to resolve a clear transition in opening of the MhOR5 receptor in which the S7b helices rotate so that a hydrophobic residue (Val468) moves out of the pore lumen and it is replaced by a polar residue (Gln467). This movement also widens the pore, to a size (9.2 Å) that can accommodate a hydrated cation. Interestingly, Gln467 is highly conserved across all insect ORs, and mutation of this residue (to alanine or arginine, but not asparagine) severely impairs function. They also identified a common binding site for two activators - eugenol and DEET - and used site-directed mutagenesis to corroborate their findings and gain further insight into ligand binding and selectivity. Their results show that the remarkably broad tuning of the receptor can be attributed to the nature of a binding site which can accommodate ligands with very different structures.

Overall this manuscript describes a significant body of work that greatly advances the field and will be of wide interest to scientists ranging from structural biologists, neuroscientists, and even ecologists. We thank the referee for their enthusiasm for this work.

-
- 1. I wonder if the authors tried mutating Val468 to glutamine (they report that mutating Val468 to asparagine is without affect). If this channel is functional, it would show that it is not just the presence of the glutamine within the pore that is important to promote ion conduction, but that the protein helix needs to rotate. (this could be concluded from the mutation to asparagine, but it is a little less direct).*

As suggested by the referee, we have now mutated Val468 to glutamine. The V468Q channel is comparable to wild-type in function, analogous to the V468N mutant, supporting that it is not simply the presence of a glutamine that is necessary to gate the pore to an open state. We have therefore replaced the V468N data with the V468Q in Figure 2 and explicitly alluded to this observation in p. 7 lines 186-188.

2. Patch-clamp data shown in Figure 1D appears to represent a single recording from either whole-cell or single-channel recording mode. It would be good to have average data (including untransfected controls).

In this revised manuscript, we significantly expanded our electrophysiological characterization of MhOR5. We performed whole-cell recordings of HEK cells transfected with MhOR5 and measured eugenol-evoked currents across a range of voltages (-80mV to 40mV). The resulting current-voltage (IV) plot reveals that, under physiological ionic conditions, the reversal potential of eugenol-evoked currents is ~0mV, consistent with MhOR5 being a non-selective cation channel. We also performed controls of untransfected HEK cells that demonstrate that eugenol does not activate any endogenous currents. Furthermore, the IV plot of untransfected HEK cells reveals that the reversal potential for these basal currents is -54mV (figure below), consistent with previous evidence that HEK cells endogenously express potassium channels (Ponce et al, *Physiol. Rep.* 2018). Together, these experiments support that MhOR5-transfected cells express a unique conductance activated by eugenol and attributable to MhOR5. Given space constraints, we simply refer to control recordings of untransfected cells as 'data not shown' in the figure legend of Fig. 1.

Reviewer Figure 1. Untransfected HEK cells do not exhibit eugenol-evoked channel activity. Left, individual traces at various voltages. Black bar indicates eugenol application. Right, IV plot of untransfected HEK cells showing a reversal potential consistent with the presence of endogenous potassium currents n=4

3. The use of one aggregate score, the "Activity index" to describe the potency of the ligands for this receptor (e.g. Fig 1c) loses essential information regarding the EC₅₀ and the maximal efficacy for the ligands. This information is plotted in the supplementary figures, so the authors might simply consider moving it to the main figures. The same is true of Fig 4 (EC₅₀ could be plotted as well as activity index).

We agree that EC₅₀, maximal efficacy, and Activity Index are complementary metrics as each reflects different aspects of how diverse odorants activate wild-type or mutant channels. We have therefore included the dose response curves for all odorants and mutants we analyzed, as well as now included a comparison of their Activity Indices and EC₅₀s. Space constraints prevent us from including all three metrics in every figure. Given

that the Activity Index incorporates both EC₅₀ and maximal efficacy, we present it in all main figures, and provide the EC₅₀ data alongside or in Extended Data Figures.

4. *It would be helpful if the authors would explain the supplementary data how MhOR5 was selected for these studies. In particular, which other receptors/species, if any, were tested.*

We focused on the receptors of *Machilis hrabei* because of the recent discovery that their genome is unique in that it contains genes for only five ORs, with no apparent gene coding for Orco. As we discuss in the maintext, this observation prompted us to reason that MhORs might serve as odorant-gated homo-tetramers. We expressed all five receptors from *Machilis hrabei* using the same protocols described in the methods section and found that only MhOR1 and MhOR5 were biochemically stable as a tetramer and amenable for further experimentation. We now include non-denaturing gels for all five MhORs in Extended Data Fig. 1, to support our rationale to focus on MhOR1 and MhOR5. These gels reveal the stoichiometry of MhORs when heterologously expressed as homomeric complexes in HEK cells and demonstrates that only MhOR1 and MhOR5 form tetramers.

Referee #2 (Remarks to the Author):

In the manuscript “The structural basis of odorant recognition in insect olfactory receptors”, del Marmol et al. present a comprehensive structural and functional study of how insect olfactory receptors recognize odorants and relay this signal to the cell. Whereas vertebrates rely on GPCR-based signalling cascades, insects have evolved odorant-gated ion channels. The main question the study wants to answer is how a rather small number of odorant receptors can recognize a very large number of different odorants by combinatorial, promiscuous binding.

The authors decided to study odorant receptors of the jumping bristletail as this organism possesses a simplified system of five receptors which combine odorant recognition and ion channeling in the same polypeptide chain. Initially, they established odorant recognition profiles of two of those five, and then continued to study the more promiscuous one, MhOR5, in detail. They obtained and compared Cryo-EM structures of MhOR5 alone and in complex with the most potent activator eugenol as well as the insect repellent DEET. Strikingly, they found that the four S7b helices of the MhOR5 tetramer which compose the central ion pore turn upon ligand binding, thereby widening the pore and exposing more hydrophilic residues which are key to ion translocation. Odorant binding appears to take place in a single hydrophobic pocket in each of the four subunits which is rather far away from the pore. By analysis of the structures, mutational studies and in silico docking experiments with a larger number of odorants, the authors establish that ligand binding does not rely on single specific interactions with certain features/chemical groups of the ligands, but rather recognizes the overall shape of the compounds by flexible hydrophobic interactions, thereby achieving the observed promiscuity.

I believe that the results presented in this manuscript will be of great interest for the broad readership of Nature as they present a big advance towards understanding the molecular basis of odorant recognition in insects. The study is well carried out, data presented in a clear and logical way, the manuscript written such that also non-experts from outside the field can immediately follow the argumentation. Hence, I would recommend to accept this manuscript for publication once the following points have been addressed by the authors.

Major points:

1. *While the conformational changes of MhOR5 upon eugenol binding are already visible in the present figures, this would be even much more obvious in one or several morph videos between the two states. Here, especially the movement of helix S7b could be shown very clearly, and it would also become even more apparent that the conformational changes close to the ligand-binding pocket are rather small.*

We now include two morph videos that link the closed and open states, viewed from the top and that side to help visualize the conformational changes during gating.

-
2. *I do not share the authors' absolute confidence in interpreting the observed lower resolution of S7b in the apo as compared to the eugenol-bound structure. First, judging only from Extended data Fig. 4, it is not completely obvious to me that the difference in local resolution of this particular helix is significantly higher than the difference in resolution over the whole molecule. Second, while I agree with the authors that it is very tempting to speculate that a lower resolution (equalling higher mobility) of the pore helix might be due to spontaneous openings of the channel, the data don't allow to draw this conclusion with certainty, and thus the authors might want to tone this part down a bit.*

We appreciate the referee's comments and agree that it is prudent to rephrase these observations to acknowledge that our interpretation is just one of multiple possible explanations. Therefore, we have now significantly tempered our statements (p. 7, lines 172-179) to make it clear that attributing the overall lower resolution of the apo structure to multiple gating states is just one possible interpretation. Nevertheless, we

now also include further evidence that lends weight to this interpretation: in Extended Data Fig. 1e, we now show that the basal calcium signal of cells transfected with MhOR5 as measured by baseline GCaMP6s fluorescence, is significantly higher than cells transfected with MhOR1 or GCaMP alone, in support of the notion that MhOR5 displays higher basal activity and transition between the open and closed state even in the absence of any added ligand.

3. *In the experiment shown in Fig. 2f, the authors compare the activation efficiency of VUAA1 on Orco with the efficiency of acetophenone on Orco+OR28. However, in order to draw firm conclusions about the effect of the Q742A mutant, the authors would have needed to use the same agonist for both complexes. Furthermore, the authors claim that the partial rescue of the effect of the Orco Q472A mutant upon co-expression with OR28 is due to a compensation of the mutant in the heteromeric assembly. However, based on the presented data and the cited references, the authors cannot be absolutely sure that both proteins form such a heteromeric complex. Even if they assemble in heteromeric complexes (and chances are high that they do), it is very likely that not a single heterotetrameric species, but rather several distinct species (4x OR, 3x OR + 1x Orco, etc) co-exist in the cells. In this situation, with the presented data the authors cannot rule out that a species consisting of 4x OR could be responsible for the observed activity and that, due to potential differences of expression levels between wt and mutant Orco, also the levels of intact 4x OR would differ, the conclusion of this experiment is overstated. Since I believe that appropriate control experiments can be done with reasonable effort, I would highly recommend that the authors perform them and present the results in a revised manuscript.*

The olfactory receptors of neopteran insects are thought to function as obligate heteromers comprised of an Orco and an OR (Larsson et al., 2004), a feature we highlight in new control experiments in which we demonstrate that OR28 alone does not function in the absence of Orco. Likewise, Orco homotetramers expressed in the absence of OR28 are not responsive to acetophenone. These controls indicate that the odorant-evoked activity of OR28 + OrcoQ472A must arise from a heteromeric complex. These controls are now included in the revised manuscript as panels in Fig. 2f.

4. *The authors present a conclusive analysis of MhOR5 and its binding specificity, but they do not provide a detailed comparison with the other MhORs or neopteran Orco or selected ORs from other species, even though they could probably at least deduce some insight into these proteins as well. Especially, it would be very interesting to know whether it is possible or even likely that these proteins harbor odorant pockets at the same position and how the volume and shape of these pockets differs from MhOR5. Even though the sequence similarity to these proteins is limited, the structures can be expected to be highly similar, so sequence alignments or homology models might give such initial insight. In the optimal case, such comparison might explain the higher selectivity of MhOR1, and even allow some predictions about the selectivity of the other three MhORs (however, this might be beyond the scope of this study).*

We thank the reviewer for this insightful suggestion, and in the revised manuscript we leverage our insights from the MhOR5 structure to explore odorant recognition in MhOR1. MhOR5 and MhOR1 display sufficient sequence conservation (36%) to allow for accurate alignment and thus identification of residues in MhOR1 homologous to those that line the binding pocket in MhOR5. We mutated 10 of these residues to alanine, replicating the mutagenesis of pocket residues we performed in MhOR5 in Fig. 3d. These functional experiments point to the fundamental structural and functional conservation of the binding pocket but also highlight potentially interesting differences that may account for the observed differences in receptor tuning.

Furthermore, we identified two residues in MhOR1 that are predicted to face away from the pocket, based on their homology to MhOR5 positions, and therefore not expected to be directly involved in ligand binding. We find that mutation of these two residues in MhOR1 has minimal impact on ligand activation (Extended Data Fig. 16), mirroring the results we obtained in MhOR5. Together, these results suggest conservation of both the overall location of the pocket as well as the orientation of the helices and residues that line it.

Additionally, we find that mutation of several MhOR1 pocket residues differentially alters their responses to 1-octanol and eugenol, with some mutants increasing the apparent affinity to one ligand while decreasing the apparent affinity to the other. This result mirrors an important observation we made based on the structure and original functional analyses of MhOR5 that mutation of individual residues lining the binding pocket alter the responses to multiple ligands, which we attribute to the fact they comprise shared structural determinants for odorant recognition.

We extend on this important observation by exploring how mutation of a single residue in the putative binding pocket of MhOR1 broadly reconfigures its tuning to multiple odorants. We found that in MhOR5, a relatively conservative substitution of Ile213 to the slightly larger residue methionine narrows its tuning to an array of odorants. In MhOR1, we identified the homologous position as Met231, mutated it to the smaller amino-acid, isoleucine, and show this mutation gives rise to slightly broader chemical tuning. Thus, exchange of residues at the same position of the binding pocket gives rise to predictable and opposing changes in chemical specificity, with MhOR5 I213M narrowing the receptor's tuning profile, while in MhOR1 M231I broadens receptor tuning. Although further structural and functional studies will undoubtedly offer a more complete picture of the distinct chemical specificity in MhOR1, these results highlight that insights obtained from MhOR5 are relevant to related members of the insect receptor family.

These results are now presented Extended Data Fig. 16 and detailed in p. 11 lines 328-346.

5. Based on the structure and alignment data, it should be possible to predict whether the formation of stable heteromeric complexes between the MhORs is likely, and it would be interesting to hear the authors' opinion on whether such complexes would be biochemically stable and physiologically relevant.

As noted above, neopteran insect olfactory receptors are thought to function as obligate heteromers with the OR co-receptor subunit (Orco). To the best of our knowledge, MhORs are the only known members of this family that can assemble as homo-tetramers, which we demonstrated as an important foundation for our subsequent structural analyses MhOR5. As such, the only available data to evaluate tetramerization comes from the present study, and while we have identified the residues involved in inter-subunit interactions in MhOR5, the pattern of sequence conservation with MhOR1 or other MhORs does not immediately reveal which features would contribute to homo-tetrameric or heteromeric assemblies. Indeed, mapping sequence conservation from the broader OR family onto the related Orco structure revealed that the highly divergent ORs each maintain a subset of residues that contribute to lining the pore and heteromeric assembly (Butterwick et al. 2018). Nevertheless, we attempted to address the essential determinants of heteromeric assembly by expressing each of the five MhORs in various combinations with each other or Orco from another species, but as of yet none of these results provided clear evidence for heterotetramer formation. We decided not to present these results in this manuscript as we believe further studies, beyond the scope of this work, would shed more light on this important issue.

Minor points:

1. The definition of the abbreviation 'ORs' is inconsistent. Once it is described as olfactory receptors (p.1 l.3) and another time as odorant receptor (p.2 l.31).

Thanks for raising this potential point of confusion. We actually use the terms odorant receptor and olfactory receptor to refer to distinct molecular species. Odorant receptor refers to the single subunit that binds odorants (an OR) whereas olfactory receptor represents the tetrameric channel.

2. *The introduction of pore architecture (p.5 l.145-148) would benefit from indicating the membrane in the corresponding Fig. 2a.*

We have noted the membrane boundaries in Fig. 2a.

3. *The authors should be more careful in their wording regarding values reported for distance of atoms (or rather atom centers) as for example on p. 5, l. 152, and the pore diameter, calculated by HOLE and displayed in Fig. 2b, that takes into account the van-der-Waals radii of the atoms.*

Thank you for the careful reading. We now explicitly state in the methods (p. 16 lines 541-546) and in the figure legend of Fig. 2 which metric is used when discussing each specific statement.

4. *P. 8, l. 242: Topologically, S5 runs parallel to S7b, but anti-parallel to S6.*

We have fixed this important structural distinction.

5. *On p.15, l. 520, a link is broken*

Thank you, we have fixed this oversight.

6. *P. 15, l. 543: 'After 30 s of a baseline recording' should be 'After 30 s of baseline recording'.*

We have changed this line as suggested.

Referee #3 (Remarks to the Author):

This manuscript by del Marmol, Yedlin and Ruta describes structural and functional analyses of an odorant receptor from the jumping bristletail, MhOR5. They find that it is a homotetrameric ion channel with high structural similarity to the Orco structure recently published by the same lab. Their three structures provide insight into the gating-related conformational changes, with a closed-pore apo structure and two open-pore liganded structures. The two ligands they use, eugenol and DEET, are quite different in chemical properties, although similar in size, and they bind to the same site, deep in the middle of each subunit. They also provide data showing that the ligand selectivity of MhOR5 is quite broad, and contrast it to that of MhOR1, which is much more selective. They also introduce structure-based mutations in MhOR5 and demonstrate that these mutations can readily tune its selectivity in a way that can be explained with the structural information.

Overall, this is a well-designed and important study, well written with a balanced discussion of the results and an effective presentation throughout. Its insights are far-reaching in terms of how we understand how Ors work, but also how odor perception works more generally. Furthermore, the insights are not limited to just the field of odor perception in insects and arthropods; many of the lessons likely extend, at least conceptually, to odor sensing by mammalian G-protein coupled receptors.

We thank the referee for their interest in this work.

I only have one significant concern with the analysis and presentation of the ligand-binding site data, which I believe can be readily addressed without significantly impacting the conclusions the authors make. I also have a few additional suggestions detailed below.

Of the mutations to the ligand binding pocket residues, the authors conclude on p. 7 that "mutation of any of these residues to alanine strongly altered eugenol signaling". While that is technically the case, the data they show do not rule out issues with protein expression, folding or trafficking, rather than alteration of the function of the ligand-binding pocket. While they do show protein expression data in Ext. Fig. 10d, it is unclear whether these strips come from a single gel image. Also, is this a western blot? The legend is unclear. Even assuming that this is a single gel image, there are variations in protein expression, and folding or trafficking were not tested. Furthermore, the methods suggest that these data are from a different protein construct (GFP-tagged) transfected using a different protocol than the one used for the functional assays. It would be much preferable to at least (1) show a single gel image with all mutants compared to WT, and (2) use the same transfection protocol as for the functional assays, even if the authors must use a different construct for western blot detection purposes.

We appreciate this important concern and now provide western blots of denaturing SDS and non-denaturing native gels in which all the mutants were run in parallel in the same experiment to allow for direct comparison (now in Extended Data Fig. 12e). Given that we rely on an anti-GFP antibody for detection on the western blot, we were unable to use the same construct for both functional experiments and these biochemical analyses but, as the referee suggests, we applied the same transfection protocol. These analyses reveal that all the mutants express at comparable levels to the wild-type channel and assemble as tetramers, in support of our conclusion that the functional deficits of these mutant receptors do not reflect expression, trafficking, or folding but disruption of odorant signaling. Moreover, a number of the mutants where odorant signaling is disrupted (e.g. F92A, S151A, I213A, or Y380A) exhibit baseline activity in the absence of ligand, as now shown in Extended Data Fig. 12a-c, further strengthening the notion that they assemble as competent ion channels but are deficient in odorant binding and/or transduction.

The other related issue I have with the data analysis is the use of the Activity Index to compare WT and mutant constructs (Figs 2e-f, 3d, 4d-e). (To be clear, I think that the Activity Index is a useful metric to compare the activity of different ligands on the same protein.) Because the Activity Index incorporates the $\Delta F/F$ ratio, it depends strongly on the basal activity of the ion channels; mutated MhOR5 with high basal activities will have

a reduced Activity Index, but that does not necessarily indicate a reduced sensitivity to the odorant. Similarly, mutated MgOR5s with lower expression levels (or trafficking to the plasma membrane) will have consequently reduced ΔF values. So, in comparing WT to mutants, the EC₅₀ values are likely the better indicators of the sensitivity to specific ligands.

We agree that the Activity Index and EC₅₀ are complementary metrics which highlight different aspects of ligand binding, but each subject to distinct limitations. The referee correctly notes that changes in baseline signaling of mutant channels may impact estimates of their Activity Indices and we therefore included measurements of EC₅₀ as a point of comparison for all mutants. We believe that the Activity Index is also a useful metric, as EC₅₀s cannot always be accurately estimated from the sub-saturating dose response curves typical of olfactory receptors, especially in the case of mutants where odorant signaling is impaired. Given this, in the revised manuscript, we include the Activity Indices, EC₅₀s, and dose-response curves for all experiments to provide the most comprehensive description of our data. It is worth noting that in Fig. 4d-e, the two binding pocket mutants, I213M and M209V, display comparable baseline activity to wild-type MhOR5, and therefore the Activity Index is likely to be an informative metric to compare changes in the tuning of these receptors. Nevertheless, we now include the EC₅₀ measured for these mutants across the entire odorant panel in Extended Data Fig. 12d.

Related to the point above, while I appreciate that baseline subtractions in Ext. Fig. 10a-c help focus the attention on the EC₅₀ changes, it would be useful to somehow indicate which mutations lead to elevated baselines (data in part (d)) in the legends to the dose-response curves.

We have now reorganized this figure (now Extended Data Fig. 12) to position the baseline fluorescence measurements next to the corresponding dose-response panels, which streamlines the reading and allows for direct comparison.

Additional specific comments and questions:

On page 6, the authors state “with Trp158 forming the lid”. This suggests that the ligand is buried and not solvent-accessible. Is that the case, and if so, is it occluded in the apo state as well? This would be worth noting in the main text. Also, in Figure 3c, is the general orientation such that the extracellular face is up? This would fit with my understanding of the “lid” description, and if that is the case, it would be worth noting in the legend.

We now provide multiple orientations of the binding pocket in a new figure, Extended Data Fig. 8, to help visualize its geometry relative to the extracellular membrane surface, displaying all the residues that we functionally characterized and color coded to match the functional measurements of Fig. 3d. We now also comment on our inability to determine the accessibility of the pocket in the apo structure due to its overall lower resolution (p. 8 lines 208-209) which we agree is an important aspect to make clear and a mechanistic question for future studies.

Related to the comment above, it would be useful to show an electrostatics surface representation of the binding pocket, to provide another illustration of its chemical properties.

We agree and have now included electrostatic surface representation of the binding pocket in Extended Data Fig. 10.

MhOR5 seems to consistently show multiple bands on SDS-PAGE western blots (Ext. Fig 1b and Ext Fig 10d). Is it because it is glycosylated? If so, are any of the sugars visible in the maps?

The extracellular loops that protrude from the micelle are generally not visible in the density, and therefore, as we note in the text, we have not built these regions of the protein. Nevertheless, it is likely that the extra bands seen in MhOR5 SDS gels reflect glycosylation, as previously reported for Orco (Lundin et al, FEBS Letters 2007).

Methods: In general, the methods are nicely detailed. The one exception I found is that the description of each construct is lacking. The plasmid constructs used for protein purification and functional assays should be detailed enough that a reader can reconstruct the exact protein sequence at a minimum (and the exact DNA sequence, ideally). As is, on p. 13, I do not know for sure the order of the Strep II tag, GFP and 3C protease sites in the construct, for example.

We have now included detailed descriptions of the constructs used throughout. In particular, the constructs used for purification are described in the first subsection of the Methods, on p. 14 lines 447-448.

On p. 14 the authors mention that caffeine docks favorably but does not activate. Does it function as a competitive inhibitor of eugenol (or DEET)?

We performed preliminary experiments to assess whether caffeine competes with eugenol by applying increasing concentrations of caffeine and a constant, near-EC₅₀ concentration of eugenol to MhOR5. We found no evidence of significant competitive antagonism from this experiment. Possible interpretations of this negative result are that caffeine does not bind to the MhOR5 pocket and its favorable docking score reflects an erroneous docking prediction, or caffeine binds but cannot engage the channel's gating mechanism. Alternatively, caffeine might be of much lower affinity than eugenol and therefore unable to displace it under the conditions of our competitive assay.

Reviewer figure 2. Increasing concentrations of caffeine are unable to outcompete eugenol in MhOR5 as assayed in the calcium influx GCaMP assay.

The validation report for the apo structure includes a resolution estimate of 3.64 Å based on the author-provided FSC curve (p. 21), rather than the 3.30 Å reported by the authors. This lower resolution seems more in line with the much lower map quality in comparison to the two liganded structures. Can the authors adjust their estimate, or justify why 3.30 Å is more appropriate? (This does not detract from the authors' conclusions, and they were very careful in their discussion of the map quality.)

As we note in the methods, the resolution estimates that we provide comes from the FSC curves that are direct output of cryoSPARC v2. Different software implementations will generally provide variable estimates,

and while we do not favor any particular source, we reason that it is best to present results from the same source for consistency. We therefore chose to report the estimate from cryoSPARC refinements for all three structures.

In Figures 1c, Ext. Fig. 1, and Ext. Fig. 2 the authors use a centered tuning curve, whereas in Figure 4e, they use a left-to-right tuning curve. I find the left-to-right presentation easier to take in. And particularly in Ext. Fig. 2 the associated dose-response panels in (d) are then organized with similarly activating ligands far apart in the figure.

We agree that the different organization of tuning curves highlights distinct aspects of receptor selectivity. We used the centered tuning curves in our initial presentation of MhOR5 and MhOR1 as this depiction highlights the breadth of their tuning and is a standard format for the characterization of olfactory receptors (e.g. Hallem and Carlson, *Cell*, 2006; Saito et al. *Science Signaling*, 2009). Importantly, in Extended Data Fig. 2 we provide the dose-response curves for all odorants, allowing readers to readily access all the experimental data. For direct comparisons between mutants in Fig. 4e and Extended Data Fig. 12d, we chose to represent the tuning curves from high-to low Activity Index, as this representation facilitates visualizing the differential changes observed across mutants and odorants.

In Figure 1e-f, the disordered loops are indicated with dotted lines. But it wasn't until I looked at the validation report that I realized that one of these loops (I believe it is the S4-S5 loop?) is 70-residues long! Looking at the two sequence alignments included in the manuscript, this loop is also long in both MhOR1 and AbOrco, but poorly conserved. A note about this loop in the section describing the overall structure would be useful.

Replacing this loop in MhOR5 with a short linker of six alanines yields fully functional channels, similar to deletion of the homologous loop in Orco. These data suggests that this loop, which is predicted to be disordered, does not have an immediately relevant functional role in odorant binding that would impact any of our conclusions. We now include these data in Extended Data Fig. 6c.

The description of the pores and their illustrations in Figure 2a-c and Ext. Fig. 12 leave me curious about the electrostatics properties of the pore surfaces, especially the side exits. Could the authors perhaps add some illustrations of the electrostatics in Ext. Fig. 12?

We thank the reviewer for this suggestion. We have now included an electrostatic surface representation of MhOR5 clipped at several planes across the protein (standard coloring, with red indicating negative potential and blue indicating positive potential), in Extended Data Fig. 10. The side exits are relatively more polar overall, as expected given that their proposed role as a conduit for ions to the intracellular solution. In addition, negatively charged residues line the side exits, in particular Glu400, which may serve to guide cations through this quadrival outlet.

In Fig. 3, it would be helpful to color-code parts (c) and (d) similarly, with the milder phenotypes (V88A, Y91A) colored differently, and if possible, include the control residues in part (c) as well.

As noted above, we now provide multiple orientations of the binding pocket in Extended Data Fig. 8 to help visualize its geometry and, as suggested by the referee, and colored coded residues to align them with the functional measurements of Fig. 3d.

Ext. Fig. 1 (and Ext. Fig. 6): I would be useful to include a panel that illustrates the white-to-blue coloring of the sequence alignments on the MhOR5 structure.

Although we agree that mapping the distribution of conservation onto the MhOR5 structure would in principle provide insight into the pattern of conservation, in the case of direct comparisons between two receptors (such as MhOR1 and MhOR5), we feel this visual representation is not informative, as the conservation is highly distributed throughout the channel (figure below), and regions of structurally and functionally relevant conservation are not evident. Mapping amino-acid conservation across a larger number receptors provides a more coherent representation of the relevant conservation patterns, as explored in Butterwick et al., Nature 2018.

Reviewer figure 3. Conservation between MhOR1 and MhOR5 mapped onto the MhOR5 structure, colored as in the alignment in Ext. Data Figure 1.

Ext. Fig. 6: what is the RMSD for the tetramer-to-tetramer superposition? And for the subunit-to-subunit superposition? (In other words, are some of the large differences in helix position at the periphery due to a shift in the tetramer organization?)

The RMSD between individual subunits is 3.00Å, and superposition of the whole tetramers yields an RMSD of 3.18Å (calculated using PDBeFold server, member of the PDB Europe). This suggests that a shift in the tetrameric organization is likely not a major contributor to the overall RMSD of these proteins. This is now included in the legend of Extended Data Figure 6.

Ext. Fig. 9: Can the authors include the chemical structures of the compounds in part (c)? (Structures of eugenol and DEET would also be useful for comparison.)

We have now added a panel in Extended Data Fig. 11 with the chemical structures of the compounds in a, b and c.

Minor notes:

On p. 15, the methods include "Error! Reference source not found."

Thank you, this has been fixed.

In at least one place (legend to Ext. Fig. 12) the authors refer to the apo structure as the “unbound structure”. It would be good to check the manuscript carefully.

We have now unified our terminology throughout.

Similarly, the methods refer to “Abak”, which I gather is A. bakeri.

Thank you, this is now fixed.

References 59 and 63 are duplicates.

Thank you for the careful reading, this has been fixed.

Reviewer Reports on the First Revision:

Referees' comments:

Referee #1 (Remarks to the Author):

This is a much-improved manuscript and the authors have addressed several of the concerns that I raised previously. I have, however, several remaining concerns (some related to new data, and others related to concerns I or another reviewer raised previously).

1. I still have concerns with respect to important patch-clamp data shown in Fig 1 that support the ion channel function of MhOR5. The authors have now added whole-cell patch-clamp data, including traces in Fig 1e and average data in Fig 1f which strengthen their conclusions. The authors now need to include the average data from untransfected controls within the manuscript, to show that the response is due to the MhOR5 channel (although I have no doubt that this is the case). Moreover, I suggest that they show average data without normalization so that the reader can know if the evoked currents are 1 pA or 1 nA, or, more likely, somewhere in between.
2. The single-channel data in Figure 1 g appears to still be from an n=1, without a control from untransfected cells. If the authors cannot report data from multiple biological replicates, and controls, this raises concerns about the reproducibility of the result and they need to leave this data out of the paper.
3. On a related point, the authors state that MhOR5 has high basal activity, but this does not appear to be the case in the new data shown in Figure 1e. If this is a “leak-subtracted trace” this author should re-do the figure without baseline subtraction, which is misleading. And if it is not subtracted, then, I would suggest limiting the discussion of spontaneous activity of MhOR5, in line with the recommendation of reviewer 2.
4. I have concerns about the tuning curves shown throughout the paper, in line with concerns raised by reviewer 3. This organization creates a misleading separation between, for example, geosmin and DEET, which have similar potency. This might be appropriate for depicting the tuning curve to light of an opsin, but it seems arbitrary when measuring ligand sensitivity, where ligands do not vary by a specific parameter. The authors responded to reviewer 3 citing literature using this measure, but that doesn't make it correct.

Referee #2 (Remarks to the Author):

The authors have addressed all points raised and in my opinion the manuscript is ready to be published.

Referee #3 (Remarks to the Author):

The original manuscript by del Marmol, Yedlin and Ruta describes structural and functional analyses of an odorant receptor from the jumping bristletail, MhOR5. They find that it is a homotetrameric ion channel with high structural similarity to the Orco structure recently published by the same lab. Their three structures provide insight into the gating-related conformational changes, with a closed-pore apo structure and two open-pore liganded structures. The two ligands they use, eugenol and DEET, are quite different in chemical properties, although similar in size, and they bind to the same site, deep in the middle of each subunit. They also provide data showing that the ligand selectivity of MhOR5 is quite broad, and contrast it to that of MhOR1, which is much more selective. They also introduce structure-based mutations in MhOR5 and demonstrate that these mutations can readily tune its selectivity in a way that can be explained with the structural information.

In the revised manuscript, the authors have reorganized the presentation of some of the data, added figures to provide a more in-depth description of the ligand-binding pocket and pore regions of MhOR5, and introduced new data from mutagenesis of the MhOR1 predicted ligand-binding pocket that provides strong additional support for their conclusions that the same ligand-binding pocket is used for recognition of many ligands, that this pocket is generally conserved in Ors, and that small alterations to the pocket can cause significant shifts in ligand selectivity.

I remain a bit skeptical about how useful the Activity Index is in comparing the response of protein variants to the same ligand (e.g. MhOR5 WT vs mutants and their response to eugenol). However, I am satisfied that the authors also provide EC50 values, EC50 curves and baseline fluorescence values, and that these data are now better organized particularly in Extended Data Fig. 12.

As I stated in my review of the original manuscript: Overall, this is a well-designed and important study, well written with a balanced discussion of the results and an effective presentation throughout. Its insights are far-reaching in terms of how we understand how Ors work, but also how odor perception works more generally. Furthermore, the insights are not limited to just the field of odor perception in insects and arthropods; many of the lessons likely extend, at least conceptually, to odor sensing by mammalian G-protein coupled receptors.

Author Rebuttals to First Revision:

The structural basis of odorant recognition in insect olfactory receptors 2021-01-01459A

Referee #1 (Remarks to the Author):

This is a much-improved manuscript and the authors have addressed several of the concerns that I raised previously. I have, however, several remaining concerns (some related to new data, and others related to concerns I or another reviewer raised previously).

We thank this reviewer for bringing these points to our attention.

1. I still have concerns with respect to important patch-clamp data shown in Fig 1 that support the ion channel function of MhOR5. The authors have now added whole-cell patch-clamp data, including traces in Fig 1e and average data in Fig 1f which strengthen their conclusions. The authors now need to include the average data from untransfected controls within the manuscript, to show that the response is due to the MhOR5 channel (although I have no doubt that this is the case). Moreover, I suggest that they show average data without normalization so that the reader can know if the evoked currents are 1 pA or 1 nA, or, more likely, somewhere in between.

We have now included average data without normalization of eugenol-evoked whole-cell currents in Extended Data Figure 2e, which demonstrates that eugenol evokes large currents (nanoAmps) in

MhOR5 transfected cells. This result is consistent with our functional imaging data, in which we observe large currents. The unnormalized data shows a range in evoked currents, as would be expected since the magnitude of current at a given voltage will be proportional to the number of channels we record from, which varies from cell to cell, due to differences in transfection efficiency. Nevertheless, this figure points to the same conclusion—that eugenol-evoked currents reverse around 0 mV in physiological conditions consistent with the idea that MhOR5 is a cation channel.

2. The single-channel data in Figure 1 g appears to still be from an n=1, without a control from untransfected cells. If the authors cannot report data from multiple biological replicates, and controls, this raises concerns about the reproducibility of the result and they need to leave this data out of the paper.

We have replaced the single-channel trace in Figure 1 to show the full time-course of eugenol application, where it can be more clearly seen that the single-channel currents begin and end with the stimulus, returning the patch to baseline activity. While this trace is representative (as noted in the figure legend), in Extended Data Figure 2e we show a family of traces from an outside-out patch recording of eugenol-evoked currents at several voltages from -60mV to +40mV. These traces replicate the properties of the whole-cell currents shown in Figure 1 but allow us to visualize single-channel fluctuations, further linking the single channel and macroscopic recordings.

3. On a related point, the authors state that MhOR5 has high basal activity, but this does not appear to be the case in the new data shown in Figure 1e. If this is a “leak-subtracted trace” this author should re-do the figure without baseline subtraction, which is misleading. And if it is not subtracted, then, I would suggest limiting the discussion of spontaneous activity of MhOR5, in line with the recommendation of reviewer 2.

We thank the reviewer for pointing out that the procedure for baseline correction was not made clear. Indeed, the whole-cell trace in Figure 1 is baseline-subtracted. HEK cells expressing MhOR5 have indeed much higher baseline activity than untransfected cells and we subtract baseline to highlight the eugenol-evoked currents, as is standard procedure (see for example, Coste et al., Science, 2010). The baseline activity of MhOR5 expressing HEK cells is extensively addressed in our imaging experiments which due to their much higher throughput nature allow us to average dozens of independent replicates using millions of cells, yielding quantitatively determined baseline activities for MhOR5 and MhOR1.

4. I have concerns about the tuning curves shown throughout the paper, in line with concerns raised by reviewer 3. This organization creates a misleading separation between, for example, geosmin and DEET, which have similar potency. This might be appropriate for depicting the tuning curve to light of an opsin, but it seems arbitrary when measuring ligand sensitivity, where ligands do not vary by a specific parameter. The authors responded to reviewer 3 citing literature using this measure, but that doesn't make it correct.

We have now also reorganized all tuning curves to display all ligands in high-to-low order, as suggested by all reviewers since it removes unnecessary arbitrariness in how the data is depicted. Our goal in using the centered tuning curves was to convey the broad sensitivity of MhOR5, using a data presentation that is more standard in describing olfactory tuning.

Referee #2 (Remarks to the Author):

The authors have addressed all points raised and in my opinion the manuscript is ready to be published.

Referee #3 (Remarks to the Author):

The original manuscript by del Marmol, Yedlin and Ruta describes structural and functional analyses of an odorant receptor from the jumping bristletail, MhOR5. They find that it is a homotetrameric ion channel with high structural similarity to the Orco structure recently published by the same lab. Their three structures provide insight into the gating-related conformational changes, with a closed-pore apo structure and two open-pore liganded structures. The two ligands they use, eugenol and DEET, are quite different in chemical properties, although similar in size, and they bind to the same site, deep in the middle of each subunit. They also provide data showing that the ligand selectivity of MhOR5 is quite broad, and contrast it to that of MhOR1, which is much more selective. They also introduce structure-based mutations in MhOR5 and demonstrate that these mutations can readily tune its selectivity in a way that can be explained with the structural information.

In the revised manuscript, the authors have reorganized the presentation of some of the data, added figures to provide a more in-depth description of the ligand-binding pocket and pore regions of MhOR5, and introduced new data from mutagenesis of the MhOR1 predicted ligand-binding pocket that provides strong additional support for their conclusions that the same ligand-binding pocket is used for recognition of many ligands, that this pocket is generally conserved in Ors, and that small alterations to the pocket can cause significant shifts in ligand selectivity.

I remain a bit skeptical about how useful the Activity Index is in comparing the response of protein variants to the same ligand (e.g. MhOR5 WT vs mutants and their response to eugenol). However, I am satisfied that the authors also provide EC50 values, EC50 curves and baseline fluorescence values, and that these data are now better organized particularly in Extended Data Fig. 12.

As I stated in my review of the original manuscript: Overall, this is a well-designed and important study, well written with a balanced discussion of the results and an effective presentation throughout. Its insights are far-reaching in terms of how we understand how Ors work, but also how odor perception works more generally. Furthermore, the insights are not limited to just the field of odor perception in insects and arthropods; many of the lessons likely extend, at least conceptually, to odor sensing by mammalian G-protein coupled receptors.